# Distinguishing examples while building concepts in hippocampal and artificial networks

Louis Kang [1,2] ✉ & Taro Toyoizumi [3,4]

The hippocampal subfield CA3 is thought to function as an auto-associative network that stores experiences as memories. Information from these experiences arrives directly from the entorhinal cortex as well as indirectly through the dentate gyrus, which performs sparsification and decorrelation. The computational purpose for these dual input pathways has not been firmly established. We model CA3 as a Hopfield-like network that stores both dense, correlated encodings and sparse, decorrelated encodings. As more memories are stored, the former merge along shared features while the latter remain distinct. We verify our model's prediction in rat CA3 place cells, which exhibit more distinct tuning during theta phases with sparser activity. Finally, we find that neural networks trained in multitask learning benefit from a loss term that promotes both correlated and decorrelated representations. Thus, the complementary encodings we have found in CA3 can provide broad computational advantages for solving complex tasks.

The hippocampus underlies our ability to form episodic memories, through which we can recount personally experienced events from our daily lives[1]. In particular, the subfield CA3 is believed to provide this capability as an autoassociative network[2–4]. Its pyramidal cells contain abundant recurrent connections exhibiting spike-timing-dependent plasticity[5,6]. These features allow networks to perform pattern completion and recover stored patterns of neural activity from noisy cues. Sensory information to be stored as memories arrives to CA3 via the entorhinal cortex (EC), which serves as the major gateway between hippocampus and neocortex (Fig. 1A). Neurons from layer II of EC project to CA3 via two different pathways[7]. First, they synapse directly onto the distal dendrites of CA3 pyramidal cells through the perforant path (PP). Second, before reaching CA3, perforant path axons branch towards the dentate gyrus (DG) and synapse onto granule cells. Granule cell axons form the mossy fibers (MF) that also synapse onto CA3 pyramidal cells, though at more proximal dendrites.

Along these pathways, information is transformed by each projection in addition to being simply relayed. DG sparsifies encodings from EC by maintaining high inhibitory tone across its numerous neurons[8]. Sparsification in feedforward networks generally decorrelates activity patterns as well[9–13]. The sparse, decorrelated nature of DG encodings is preserved by the MF pathway because its connectivity is also sparse; each CA3 pyramidal cell receives input from only ≈50 granule cells[14]. In contrast, PP connectivity is dense with each CA3 pyramidal cell receiving input from ≈4000 EC neurons[14], so natural correlations between similar sensory stimuli should be preserved. Thus, CA3 appears to receive two encodings of the same sensory information with different properties: one sparse and decorrelated through MF and the other dense and correlated through PP. What is the computational purpose of this dual-input architecture? Previous theories have proposed that the MF pathway is crucial for pattern separation during memory storage, but retrieval is predominantly mediated by the PP pathway and can even be hindered by MF inputs[15–17]. In these models, MF and PP encodings merge during storage and one hybrid pattern per memory is recovered during retrieval.

[1]Neural Circuits and Computations Unit, RIKEN Center for Brain Science, 2-1 Hirosawa, Wako-shi, Saitama 351-0198, Japan. [2]Graduate School of Informatics, Kyoto University, 36-1 Yoshida-honmachi, Sakyo-ku, Kyoto 606-8501, Japan. [3]Laboratory for Neural Computation and Adaptation, RIKEN Center for Brain Science, 2-1 Hirosawa, Wako-shi, Saitama 351-0198, Japan. [4]Graduate School of Information Science and Technology, University of Tokyo, 7-3-1 Hongo, Bunkyo-ku, Tokyo 113-0033, Japan. ✉e-mail: louis.kang@riken.jp

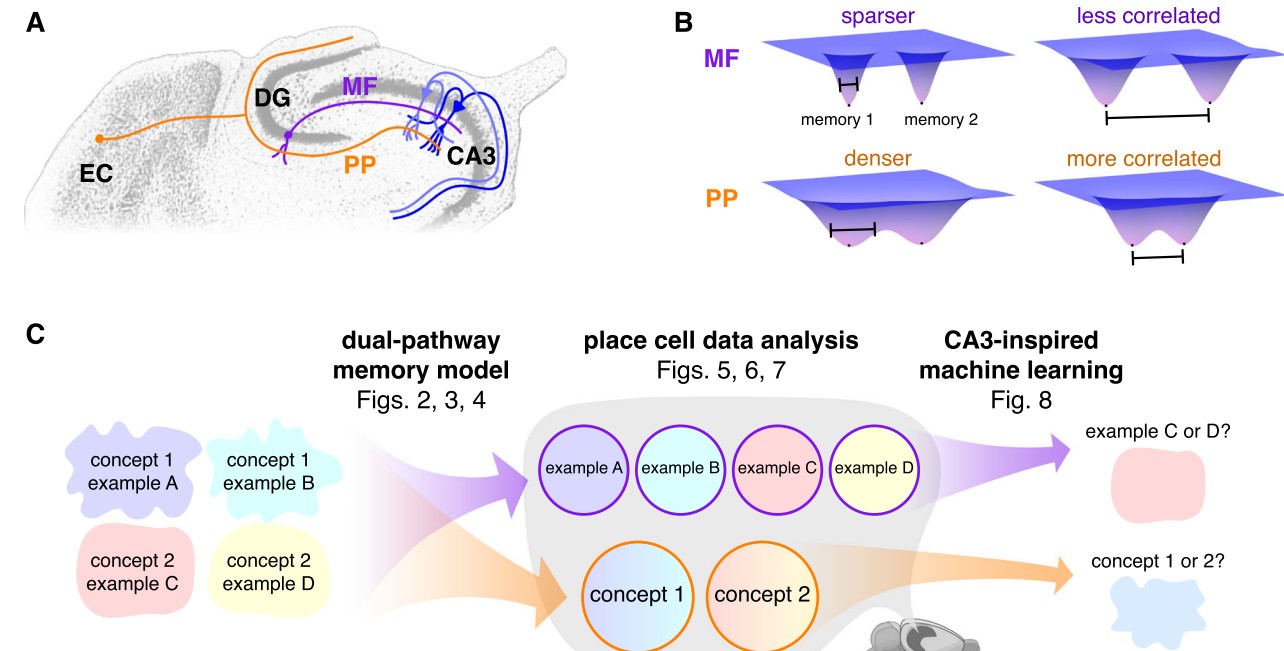

**Fig. 1 | Overview and motivation. A** Entorhinal cortex (EC) projects to CA3 directly via the perforant path (PP, orange) as well as indirectly through the dentate gyrus (DG) via mossy fibers (MF, purple). Adapted from Rosen GD, Williams AG, Capra JA, Connolly MT, Cruz B, Lu L, Airey DC, Kulkarni K, Williams RW (2000) The Mouse Brain Library @ https://www.mbl.org. Int Mouse Genome Conference 14: 166. https://www.mbl.org. **B** MF memory encodings are believed to be sparser and less correlated compared to PP encodings. In an autoassociative network, attractor basins of the former tend to remain separate and those of the latter tend to merge.

**C** By modeling hippocampal networks, we first predict that MF and PP encodings in CA3 can respectively maintain distinctions between memories and generalize across them (Figs. 2–4). By analyzing publicly available neural recordings, we then detect signatures of these encoding properties in rat CA3 place cells (Figs. 5–7). By training artificial neural networks, we finally demonstrate that these encoding types are suited to perform the complementary tasks of example discrimination and concept classification (Fig. 8).

Instead, we consider the possibility that CA3 can store both MF and PP encodings for each memory and retrieve either of them. Inhibitory tone selects between the two; with a higher activity threshold, sparser MF patterns are more likely to be recovered, and the opposite holds for denser PP patterns. By encoding the same memory in two different ways, each can be leveraged for a different computational purpose. Conceptually, in terms of energy landscapes, sparser patterns have narrower attractor basins than denser patterns because fewer neurons actively participate (Fig. 1B). Moreover, less correlated patterns are located farther apart compared to more correlated patterns. Thus, MF energy basins tend to remain separate with barriers between them, a property called pattern separation that maintains distinctions between similar memories and is known to exist in DG[18–20]. In contrast, PP energy basins tend to merge, which enables the clustering of similar memories into concepts. This proposed ability for CA3 to recall both individual experiences and generalizations across them would explain observed features of hippocampal function. For instance, remembering the details of a recent visit with an acquaintance is an example of hippocampus-dependent episodic memory[1,21]. Meanwhile, hippocampal neurons can also generalize over your visits and respond to many different representations of your acquaintance, including previously unseen photographs or her name in spoken or written form[22,23].

To instantiate these ideas, we constructed a model for EC, DG, and CA3 in which CA3 stores both MF and PP encodings of each memory (Fig. 1C). We observe that MF encodings remain distinct, whereas PP encodings perform concept learning by merging similar memories. Our model predicts relationships between coding properties and network sparsity across phases of the theta oscillation, which modulates inhibitory tone in the hippocampal region. We tested these predictions across two publicly available datasets[24,25],

and each analysis reveals that tuning of CA3 neurons is sharper during sparse theta phases and broader during dense phases. This supports our model and enriches our understanding of phase coding in hippocampus. While our model does not include CA1, we present comparative experimental analyses for this subfield in various Supplementary Figures. Beyond asserting the presence of complementary encodings in CA3, we demonstrate that they can offer functional advantages. Applying inspiration from our CA3 model and data analysis toward machine learning, we introduced a plug-and-play loss function that endows artificial neural networks with both correlated, PP-like and decorrelated, MF-like representations. These networks can perform better in multitask learning compared to networks with single representation types, which suggests a promising strategy for helping neural networks to solve complex tasks. While the essential components of our networks are explained in the Results section, their full descriptions and justifications are provided in the Methods section with parameter values in Table 1.

## Results

### MF encodings remain distinct while PP encodings build concepts in our model for CA3

We model how representations of memories are transformed along the two pathways from EC to CA3 and then how the resultant encodings are stored and retrieved in CA3. First, we focus on the transformations between memories and their CA3 encodings. The sensory inputs whose encodings serve as memories in our model are FashionMNIST images[26], each of which is an example belonging to one of three concepts: sneakers, trousers, and coats (Fig. 2A). They are converted to neural activity patterns along each projection from EC to CA3 (Fig. 2B). Our neurons are binary with activity values of 0 or 1. Each image $\mathbf{i}_{\mu\nu}$ representing example $\nu$ in concept $\mu$ is first encoded by EC

using a binary autoencoder, whose middle hidden layer activations represent the patterns $\mathbf{x}_{\mu\nu}^{EC}$ (Fig. 2C). Only 10% of the neurons are allowed to be active, so the representation is sparse, and there are more EC neurons than image pixels, so the representation is over-complete; sparse, overcomplete encoding models and autoencoder

### Table 1 | Key hippocampus model parameters and their values unless otherwise noted

| Parameter | Value | See also |
|---|---|---|
| number of concepts | 3 | Fig. 2A |
| examples stored per concept $s$ | 1–100 | Fig. 3 |
| EC network size $N_{EC}$ | 1024 | Eqs. (5) and (6), Fig. 2C, D |
| DG network size $N_{DG}$ | 8192 | Eq. (6), Fig. 2D |
| CA3 network size $N_{CA3}$ | 2048 | Eqs. (6) and (8), Figs. 2D and 3 |
| EC pattern density $a_{EC}$ | 0.1 | Eqs. (5) and (6), Fig. 2C, D |
| DG pattern density $a_{DG}$ | 0.005 | Eq. (6), Fig. 2D |
| MF pattern density $a_{MF}$ | 0.02 | Eq. (6), Fig. 2D |
| PP pattern density $a_{PP}$ | 0.2 | Eq. (6), Fig. 2D |
| EC pattern correlation $\rho_{EC}$ | 0.15 | Eqs. (5) and (6), Fig. 2C, D |
| DG pattern correlation $\rho_{DG}$ | 0.02 | Eq. (6), Fig. 2D |
| MF pattern correlation $\rho_{MF}$ | 0.01 | Eq. (6), Fig. 2D |
| PP pattern correlation $\rho_{PP}$ | 0.09 | Eq. (6), Fig. 2D |
| PP pattern strength $\zeta$ | 0.1 | Eq. (8), Fig. 3A |
| fraction of neurons flipped to form cue | 0.01 | Fig. 3B |
| rescaled threshold $\theta'$ | 0–0.5 | Eq. (11), Fig. 3C |
| rescaled inverse temperature $\beta'$ | 100 | Eq. (11) |

neural networks are common unsupervised models for natural image processing[27–30].

From EC, we produce DG, MF, and PP encodings with random, binary, and sparse connectivity matrices between presynaptic and postsynaptic regions (Fig. 2D), i.e., from EC to DG, from DG to CA3 via MF, and from EC to CA3 via PP. Each matrix transforms presynaptic patterns $\mathbf{x}_{\mu\nu}^{pre}$ into postsynaptic inputs, which are converted into postsynaptic patterns $\mathbf{x}_{\mu\nu}^{post}$ at a desired density using a winners-take-all approach. That is, the postsynaptic neurons receiving the largest inputs are set to 1 and the others are set to 0. We define *density* to be the fraction of active neurons, so lower values correspond to sparser patterns. Enforcing a desired postsynaptic pattern density is equivalent to adjusting an activity threshold. At CA3, two encodings for each image converge: $\mathbf{x}_{\mu\nu}^{MF}$ with density 0.02 and $\mathbf{x}_{\mu\nu}^{PP}$ with density 0.2. Not only are MF patterns sparser, they are less correlated with average correlation 0.01, compared to a corresponding value of 0.09 for PP patterns. Such an association between sparsification and decorrelation has been widely reported across many theoretical models and brain regions[10–13], and it is also captured by our model. Decreasing postsynaptic pattern density (sparsification) correspondingly decreases the postsynaptic correlation (decorrelation) for any presynaptic statistics (Fig. 2E). We contribute further insight by deriving an explicit mathematical formula that connects densities and correlations of patterns in presynaptic and postsynaptic networks:

$$\rho_{post} = \frac{\Gamma\left[\sqrt{2}\,\mathrm{erfc}^{-1}(2a_{post}), a_{pre} + \rho_{pre} - a_{pre}\rho_{pre}\right] - a_{post}^2}{a_{post}(1 - a_{post})},$$

$$\text{where} \quad \Gamma[\phi,\sigma] \equiv \frac{1}{2\pi} \int_{\arccos\sigma}^{\pi} d\psi \, \exp\left[-\frac{\phi^2}{1 + \cos\psi}\right] \tag{1}$$

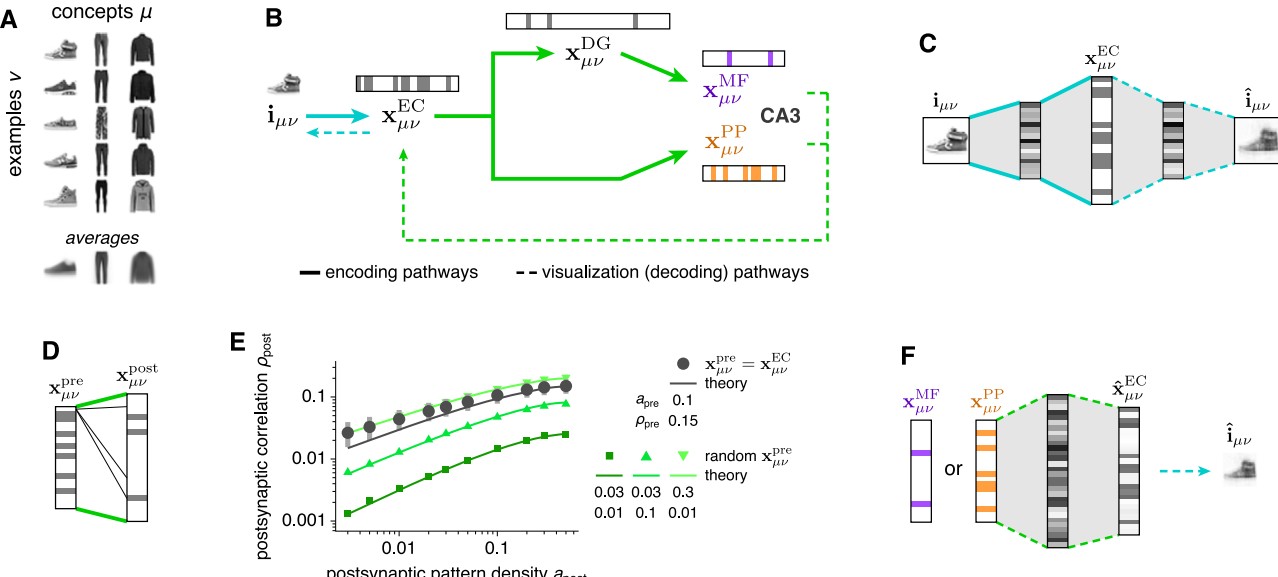

**Fig. 2 | We model the transformation of memory representations along hippocampal pathways; MF and PP encodings of the same memories converge at CA3. A** Memories are FashionMNIST images, each of which is an example of a concept. **B** Overview of model pathways. Encoding pathways correspond to the biological architecture in Fig. 1A. Decoding pathways are used to visualize CA3 activity and are not intended to have biological significance. **C** We use an auto-encoder with a binary middle layer to transform each memory $\mathbf{i}_{\mu\nu}$ into an EC pattern $\mathbf{x}_{\mu\nu}^{EC}$. **D** From EC to CA3, we use random binary connectivity matrices to transform each presynaptic pattern $\mathbf{x}_{\mu\nu}^{pre}$ to a postsynaptic pattern $\mathbf{x}_{\mu\nu}^{post}$. **E** Enforcing sparser postsynaptic patterns in **D** promotes decorrelation. Dark gray indicates use of $\mathbf{x}_{\mu\nu}^{EC}$ as presynaptic patterns. Points indicate means and bars indicate standard deviations over 8 random connectivity matrices. Green indicates randomly generated presynaptic patterns at various densities $a_{pre}$ and correlations $\rho_{pre}$. Theoretical curves depict Eq. (1). **F** To visualize CA3 encodings, we pass them through a feedforward network trained to produce the corresponding $\mathbf{x}_{\mu\nu}^{EC}$ for each $\mathbf{x}_{\mu\nu}^{MF}$ and $\mathbf{x}_{\mu\nu}^{PP}$. Images are then decoded using the autoencoder in **C**. Source data are provided as a Source Data file.

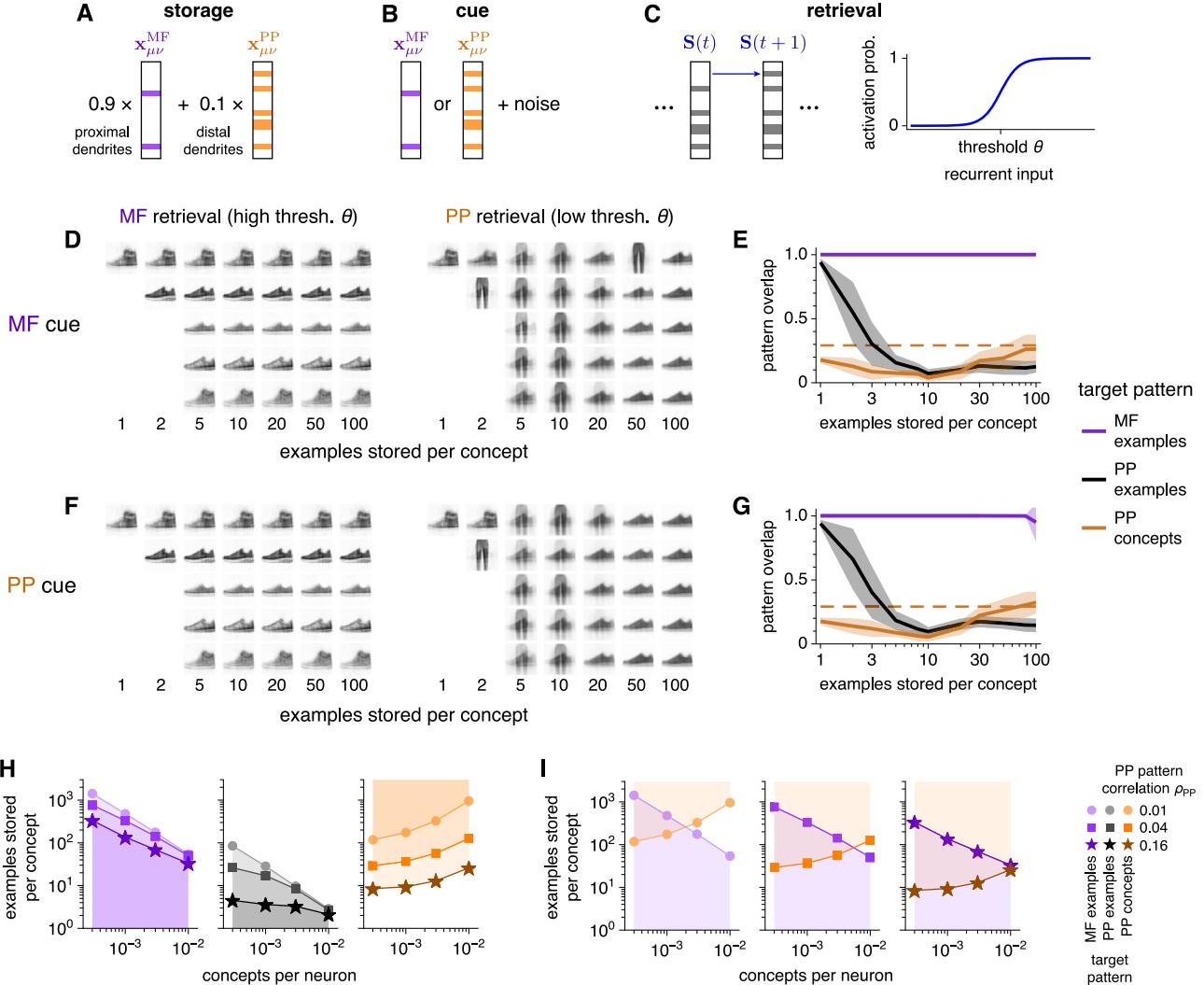

**Fig. 3 | We model CA3 to store both MF and PP encodings of the same memories; MF examples remain distinct while PP examples build concept representations. A**–**C** Overview of the Hopfield-like model for CA3. **A** We store linear combinations of MF and PP encodings, with greater weight on the former because MF inputs are stronger. **B** Retrieval begins by initializing the network to a stored pattern corrupted by flipping the activity of randomly chosen neurons. **C** During retrieval, the network is asynchronously updated with a threshold $\theta$ that controls the desired sparsity of the recalled pattern. **D**, **E** Retrieval behavior using MF cues. Examples from the three concepts depicted in Fig. 2A are stored. **D** Visualizations of retrieved patterns. MF encodings, retrieved at high $\theta$, maintain distinct representations of examples. PP encodings, retrieved at low $\theta$, merge into concept representations as more examples are stored (compare with average image in

Fig. 2A). **E** Overlap of retrieved patterns with target patterns: MF examples, PP examples, or PP concepts defined by averaging over PP examples and binarizing (Methods). Solid lines indicate means, shaded regions indicate standard deviations, and the dashed orange line indicates the theoretically estimated maximum value for concept retrieval (Methods). In all networks, up to 30 cues are tested. **F, G** Similar to **D**, **E**, but using PP cues. **H** Network capacities computed using random MF and PP patterns instead of FashionMNIST encodings. Shaded regions indicate regimes of high overlap between retrieved patterns and target patterns (Supplementary Information). MF patterns have density 0.01 and correlation 0. PP patterns have density 0.5. **I** Similar to **H**, but overlaying capacities for MF examples and PP concepts to highlight the existence of regimes in which both can be recovered. Source data are provided as a Source Data file.

and erfc$^{-1}$ is the inverse complementary error function. In other words, given the density $a_{\text{pre}}$ and correlation $\rho_{\text{pre}}$ of the presynaptic patterns and the desired density $a_{\text{post}}$ of the postsynaptic patterns, the postsynaptic correlation $\rho_{\text{post}}$ is determined. Equation (1) is remarkable in that only these four quantities are involved, revealing that at least in some classes of feedforward networks, other parameters such as network sizes, synaptic density, and absolute threshold values do not contribute to decorrelation. It is derived in Supplementary Information, and its behavior is further depicted in Supplementary Fig. 2B, C.

Ultimately, the encoding pathways in Fig. 2C–E provide CA3 with a sparse, decorrelated $\mathbf{x}_{\mu\nu}^{\text{MF}}$ and a dense, correlated $\mathbf{x}_{\mu\nu}^{\text{PP}}$ for each memory, in accordance with our biological understanding (Fig. 1A, B). Next, we aim to store these patterns in an autoassociative model of CA3. Before doing so, we develop visualization pathways that decode CA3

representations back into images, so memory retrieval can be intuitively evaluated. This is accomplished by training a continuous-valued feedforward network to associate each MF and PP pattern with its corresponding EC pattern (Fig. 2F). From there, the reconstructed EC pattern can be fed into the decoding half of the autoencoder in Fig. 2C to recover the image encoded by CA3. These decoding pathways are for visualization only and are not designed to mimic biology, although there may be parallels with the neocortical output pathway from CA3 to CA1 and deep layers of EC[7]. The neuroanatomical connectivity of CA1 is more complex and includes temporoammonic inputs from EC as well as strong secondary outputs through the subiculum, which also reciprocally connects with EC.

Now, we model memory storage in the CA3 autoassociative network. For each example $\nu$ in concept $\mu$, its MF encoding $\mathbf{x}_{\mu\nu}^{\text{MF}}$ arrives at

the proximal dendrites and its PP encoding $\mathbf{x}_{\mu\nu}^{\mathrm{PP}}$ arrives at the distal dentrites of CA3 pyramidal cells (Fig. 3A). The relative strength of PP inputs is weaker because PP synapses are located more distally and are observed to be much weaker than MF synapses, which are even called detonator synapses[7,31,32]. The inputs are linearly summed and stored in a Hopfield-like network[33], with connectivity

$$W_{ij} \sim \sum_{\mu\nu} \left( 0.9\, x_{\mu\nu i}^{\mathrm{MF}} + 0.1\, x_{\mu\nu i}^{\mathrm{PP}} \right) \left( 0.9\, x_{\mu\nu j}^{\mathrm{MF}} + 0.1\, x_{\mu\nu j}^{\mathrm{PP}} \right), \qquad (2)$$

where $i$ and $j$ are respectively postsynaptic and presynaptic neurons. Equation (2) captures the most crucial terms in $W_{ij}$; see Methods for the full expression. While we assume linear summation between $\mathbf{x}_{\mu\nu}^{\mathrm{MF}}$ and $\mathbf{x}_{\mu\nu}^{\mathrm{PP}}$ for simplicity, integration of inputs across CA3 dendritic compartments is known to be nonlinear[17,34,35]. Moreover, sublinear summation can also arise from a temporal offset between MF and PP inputs, in which case changes in synaptic weights across pathways could be weaker than those within the same pathway according to spike-timing-dependent plasticity[5,6]. In Supplementary Fig. 3F, we show that network behavior can be maintained when nonlinearity is introduced.

In previous models, CA3 would retrieve only MF encodings, only PP encodings, or only the activity common between MF–PP pairs[15–17]. We assess the ability of the network to retrieve either $\mathbf{x}_{\mu\nu}^{\mathrm{MF}}$ or $\mathbf{x}_{\mu\nu}^{\mathrm{PP}}$ using either encoding as a cue (Fig. 3B). Each cue is corrupted by flipping randomly chosen neurons between active and inactive and is set as the initial network activity. During retrieval, the network is asynchronously updated via Glauber dynamics[36]. That is, at each simulation timestep, one neuron is randomly selected to be updated (Fig. 3C). If its total input from other neurons exceeds a threshold $\theta$, then it is more likely to become active; conversely, subthreshold total input makes silence more likely. The width of the sigmoid function in Fig. 3C determines the softness of the threshold. A large width implies that activation and silence are almost equally likely for recurrent input near threshold. A small width implies that activation is almost guaranteed for recurrent input above threshold and almost impossible for input below threshold. See Methods for the full expression of this update rule.

The threshold $\theta$ represents the general inhibitory tone of CA3 and plays a key role in retrieval. At high $\theta$, neural activity is disfavored, so we expect the network to retrieve the sparser, more strongly stored MF encoding of the cue. Upon lowering $\theta$, more neurons are permitted to activate, so those participating in the denser, more weakly stored PP encoding should become active as well. Because our neurons are binary, active neurons in either the MF or the PP encoding would have the same activity level of 1, even though their connectivity strengths differ. This combined activity of both encodings is almost the same as the PP encoding alone, which contains many more active neurons. Thus, we expect the network to approximately retrieve the PP encoding at low $\theta$.

Figure 3 D–G illustrates the central behavior of our CA3 model; see Supplementary Fig. 3A, B for trouser and coat visualizations, which behave similarly to the sneaker visualizations shown here. First using MF encodings as cues, we seek to retrieve either MF or PP encodings by respectively setting a high or low threshold. As we load the network with increasingly more stored examples, distinct MF examples can consistently be retrieved with high threshold (Fig. 3D). Meanwhile, retrieval of PP examples with low threshold fails above 1–2 examples stored per concept. At large example loads, the network again retrieves a sneaker memory when cued with sneaker examples. However, this memory is the same for all sneaker cues and appears similar to the average image over all sneaker examples (Fig. 2A), which captures common sneaker features (Supplementary Fig. 2A). Thus, the network is retrieving a representation of the sneaker *concept*. Notably, concepts are never directly presented to the network; instead, the

network builds them through the unsupervised accumulation of correlated examples. The retrieval properties visualized in Fig. 3D are quantified in Fig. 3E by computing the overlap between retrieved and target patterns. Across all example loads shown, retrieved MF patterns overlap with target examples. As example load increases, retrieved PP patterns transition from encoding examples to representing concepts. We define the target pattern $\mathbf{x}_{\mu}^{\mathrm{PP}}$ for a PP concept $\mu$ by activating the most active neurons across PP examples within that concept until the PP density is reached, and the dashed line in Fig. 3E coarsely estimates the largest overlap achievable (Methods).

The network capabilities observed for MF cues are preserved when we instead use PP cues (Fig. 3F, G) or cues combining the neurons active in either encoding (Supplementary Fig. 3C–E); again, these latter two are similar because MF encodings are sparse. Thus, retrieval behavior is driven largely by the level of inhibition rather than the encoding type of cues. This feature implies that our model is agnostic to whether memory retrieval in the hippocampus is mediated by the MF pathway, the PP pathway, or both. Computationally, it implies that our model can not only retrieve two encodings for each memory but also perform heteroassociation between them. Autoassociation and heteroassociation are preserved over large ranges in model parameters (Supplementary Fig. 3F).

To show that concept target patterns $\mathbf{x}_{\mu}^{\mathrm{PP}}$ and average images within concepts are indeed valid representations of concepts for our image dataset, we plot them in image space after transforming $\mathbf{x}_{\mu}^{\mathrm{PP}}$ through the visualization pathway (Supplementary Fig. 2A). We observe that these two representation types appear similar to each other and lie near the centers of well-separated clusters of examples for each concept. In machine learning, clustering around central points is a common paradigm for unsupervised category learning, with $k$-means clustering as an example. In cognitive science as well, clustering has been used as model for unsupervised category learning[37,38], and central representations called prototypes can be used for category assignment[39]. Thus, we conclude that $\mathbf{x}_{\mu}^{\mathrm{PP}}$ and average images can indeed serve as concept representations. With more complex image datasets, such as CIFAR10[40], examples may not be clustered in image space or in encoding space with our model's simple autoencoder. To learn concepts, nonlinear decision boundaries can identified using supervised algorithms, but these complicated partitions of space may not admit central prototypes that accurately represent concepts. Alternatively, we can employ more sophisticated feature extraction techniques to map examples into an encoding space that exhibits clusters with simple boundaries between concepts. If that is achieved, then central features such as averages within clusters in that space can again serve as concept representations. More powerful feature extraction can be incorporated in our model by substituting our simple autoencoder with, for example, an unsupervised variational autoencoder or a supervised deep classifier.

We investigate retrieval more comprehensively by randomly generating MF and PP patterns across a broader range of statistics instead of propagating images along the hippocampal pathways in Fig. 2B (Methods). For simplicity, we take MF examples to be uncorrelated. In Fig. 3H, I, we show regimes for successful retrieval of MF examples, PP examples, and PP concepts. For MF and PP examples, the network has a capacity for stored patterns above which they can no longer be retrieved (Fig. 3H). For PP concepts, the network requires storage of a minimum number of examples below which concepts cannot be built. As expected intuitively, fewer examples are needed if they are more correlated, since common features can be more easily deduced. Figure 3I overlays retrieval regimes for MF examples and PP concepts. When the number of concepts is low, there exists a regime at intermediate numbers of stored examples in which both examples and concepts can be retrieved. This multiscale retrieval regime corresponds to the network behavior observed in Fig. 3D–G, and it is larger for more correlated PP

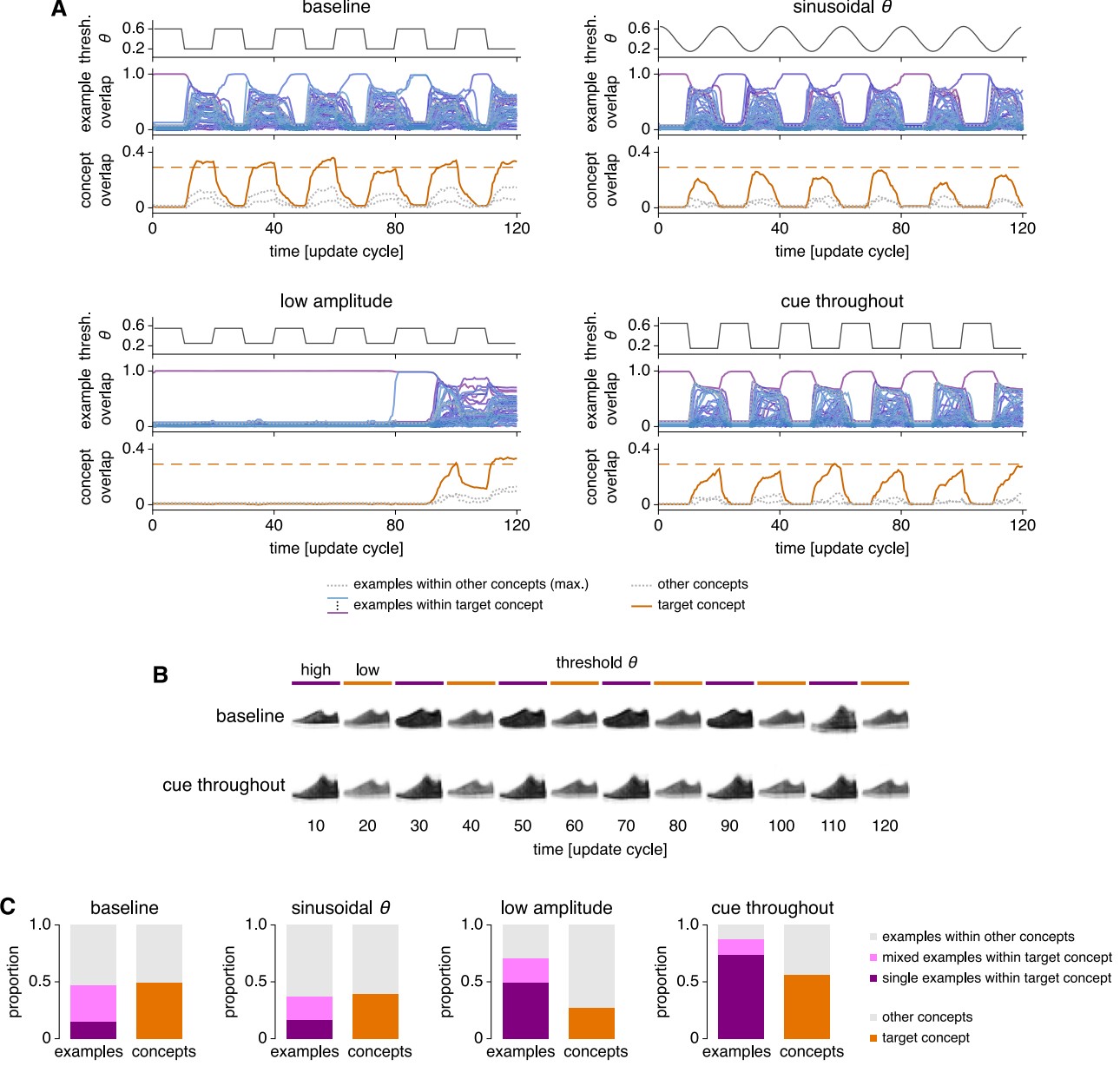

**Fig. 4 | The CA3 model can alternate between MF example and PP concept representations under an oscillating threshold.** Four scenarios are considered: a baseline condition with abrupt threshold changes, sinusoidal threshold changes, threshold values of 0.55 and 0.25 instead of 0.6 and 0.2, and the weak input of an MF cue throughout the simulation instead of only at the beginning. **A** Pattern overlap dynamics. Each panel shows, from top to bottom, the threshold, overlaps with MF examples, and overlaps with PP concepts. The dashed orange line indicates the theoretically estimated maximum value for concept retrieval (Methods). **B** Visualizations of retrieved patterns show alternation between examples and encodings. On the other hand, its size does not substantially change with the sparsity of MF patterns (Supplementary Fig. 3G, H). Our capacity values agree with theoretical formulas calculated using techniques from statistical physics[41]. In all, our networks with randomly generated patterns demonstrate that our results generalize to larger networks that store more examples in more concepts and are not idiosyncratic to the pattern generation process in Fig. 2.

concepts. In the baseline case, various examples are explored; in the cue-throughout case, the same cued example persists. **C** Summary of retrieval behavior between update cycles 60 to 120. For each scenario, 20 cues are tested in each of 20 networks. Each panel depicts the fractions of simulations demonstrating various example (left) and concept (right) behaviors. In all networks, 50 randomly chosen examples from each of the 3 concepts depicted in Fig. 2A are stored. One update cycle corresponds to the updating of every neuron in the network (Methods). Source data are provided as a Source Data file.

To further explore the heteroassociative capability of our network, we cue the network with an MF pattern and apply a time-varying threshold during retrieval. The network representation can then alternate between the PP concept of the original cue during oscillation phases with low threshold and various MF examples of that concept during phases with high threshold (Fig. 4A, B). Sharply and sinusoidally varying threshold values both produce this behavior. From one oscillation cycle to the next, the MF encoding can hop among different examples because concept information is preferentially preserved over example information during low-threshold phases. If we weakly apply the MF cue as additional neural input throughout the simulation (Methods), the network will only alternate between the target MF example and the target PP concept. This condition can represent memory retrieval with ongoing sensory input. If we decrease the amplitude of the oscillation,

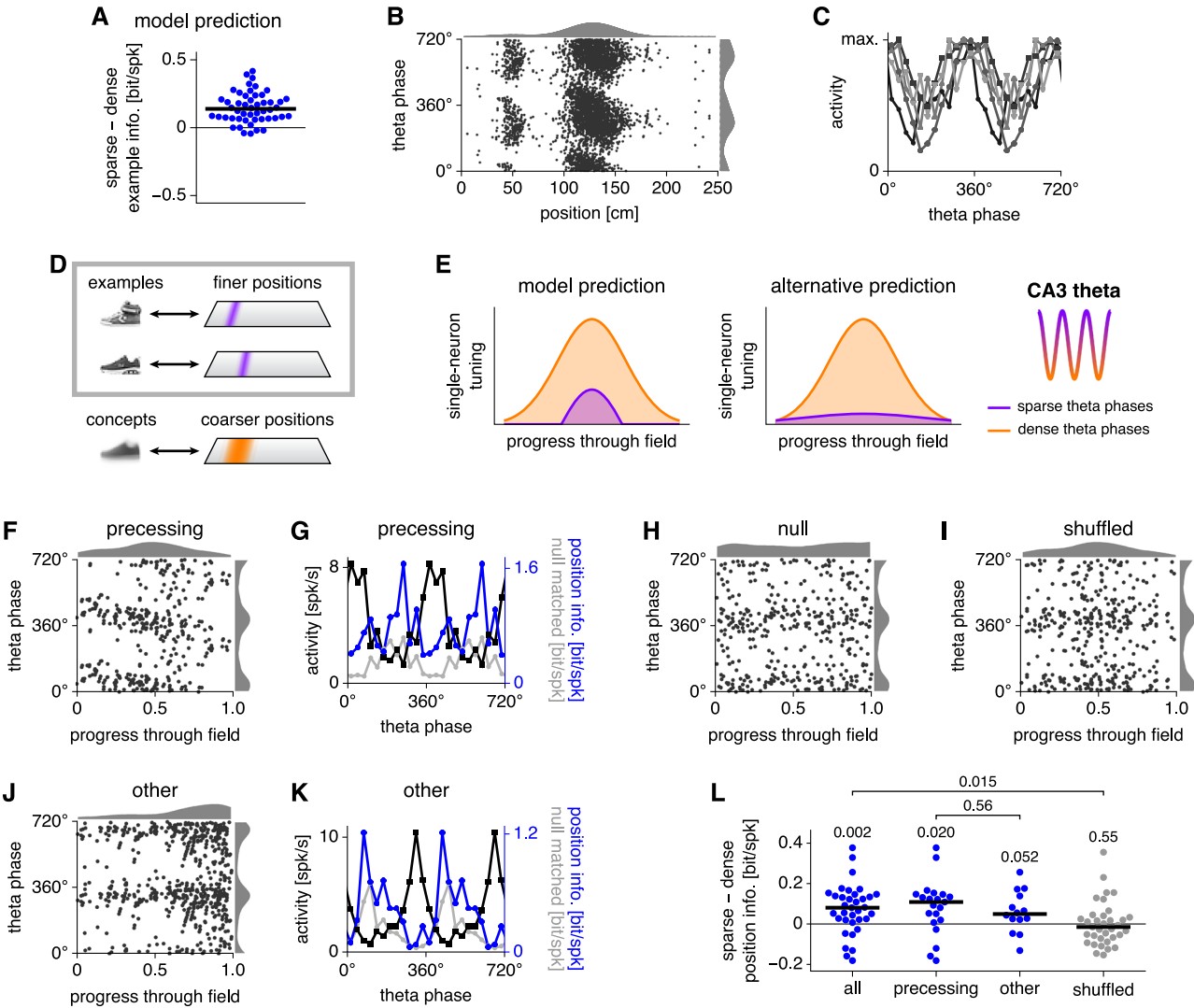

**Fig. 5 | Place field data support the model prediction that sparser theta phases should preferentially encode finer, example-like positions. A** Our CA3 model predicts that single neurons convey more information per spike about example identity during sparse regimes. Each point represents a neuron, n = 50. **B** Example CA3 place cell activity along a linear track. Each spike is represented by two points at equivalent phases with histograms over position (top) and over theta phase (right). **C** Activity by theta phase for 5 CA3 place cells. **D** To test our model, we construe CA3 place cells to store fine positions as examples, which can combine into coarser regions as concepts. Here, we focus on example encoding. **E** Our model predicts that CA3 place fields are more sharply tuned during sparse theta phases. An alternative hypothesis is sharper tuning during dense phases. **F** Example phase-precessing place field. **G** Activity (black), raw position information per spike (blue), and mean null-matched position information (gray) by theta phase for the field in **F**.

Sparsity-corrected position information is the difference between the raw and mean null-matched values. **H** Null-matched place field obtained by replacing spike positions, but not phases, with uniformly distributed random values. **I** Shuffled place field obtained by permuting spike phases and positions. **J, K** Similar to **F, G**, but for a place field that is not precessing. **L** Average difference in position information between the sparsest and densest halves of theta phases. For all cell populations, sparse phases convey more position information per spike. Each point represents a field. All and shuffled n = 35, precessing n = 21, and other n = 14. Numbers indicate *p*-values calculated by two-tailed Wilcoxon signed-rank tests except for the comparison between precessing and other, which is calculated by the two-tailed Mann-Whitney *U* test. For all results, spikes during each traveling direction are separately analyzed. In **A** and **L**, information is sparsity-corrected with horizontal lines indicating medians. Source data are provided as a Source Data file.

alternation between examples and concepts is disrupted and the network favors one encoding type over the other. We quantify the distribution of network behaviors during high- and low-threshold phases in Fig. 4C. The proportion of simulations in which single MF patterns are retrieved, the persistence of the target PP concept, and other retrieval properties vary with network parameters. In Supplementary Fig. 4A, B, we present analogous results for randomly generated MF and PP patterns demonstrating that these retrieval properties also depend on MF pattern sparsity. All in all, while our network can represent either examples or concepts at each moment in time, an oscillating threshold provides access to a range of representations over every oscillation cycle.

## Place cell data reveals predicted relationships between encoding properties and theta phase

The central feature of our CA3 model is that an activity threshold determines whether the network retrieves example or concept encodings. We claim that the theta oscillation in CA3 physiologically implements this threshold and drives changes in memory scale. To be specific, our model predicts that single neurons should convey more information per spike about example identity during epochs of sparser activity (Fig. 5A). This single-neuron prediction can be tested by analyzing publicly available datasets of CA3 place cells. Figure 5B shows one example place cell recorded while a rat traverses a linear track[24,42]. During locomotion, single-neuron activity in CA3 is strongly

modulated by the theta oscillation (Fig. 5C); we use this activity as an indicator of network sparsity since a relationship between the two has been observed (Fig. 3 in Skaggs et al.[43]). We assume an equivalence between the encoding of images by our CA3 model and the encoding of spatial positions by CA3 place cells (Fig. 5D). Examples are equivalent to fine positions along the linear track. Just as similar examples merge into concepts, nearby positions can aggregate into coarser regions of space. Through this equivalence, we can translate the prediction about example information per spike (Fig. 5A) into a prediction about spatial tuning (Fig. 5E). During denser theta phases, place fields should be broader, which corresponds to lower position information per spike. This prediction relies on our claim that the theta oscillation in CA3 acts as the inhibitory threshold of our model. A priori, the alternative prediction that place fields are sharper during dense theta phases is an equally valid hypothesis. Higher activity may result from strong drive by external stimuli that the neuron serves to encode, while lower activity may reflect noise unrelated to neural tuning. The sharpening of visual tuning curves by attention is an example of this alternative prediction[44]. From a more general perspective, the model and alternative predictions in Fig. 5E roughly correspond to subtractive and divisive modulation of firing rates, respectively. Both kinds of inhibitory effects are found in cortical circuits[45–47]. We will now test whether experimental data reflect our model prediction of sharper place field tuning with higher spatial information during sparser theta phases, which would support a subtractive role of theta as an oscillating inhibitory threshold over a divisive one.

First, we investigate the encoding of fine, example-like positions by analyzing phase-dependent tuning within single place fields. We use the Collaborative Research in Computational Neuroscience (CRCNS) hc-3 dataset contributed by György Buzsáki and colleagues[24,42]. Figure 5F shows one extracted field that exhibits phase precession (for others, see Supplementary Fig. 5A). At each phase, we compute the total activity as well as the information per spike conveyed about position within the field (Fig. 5G and Supplementary Fig. 5B). It is well known that the estimation of information per spike is strongly biased by sparsity. Consider the null data in Fig. 5H that is matched in spike phases; spike positions, however, are randomly chosen from a uniform distribution. In the large spike count limit, uniformly distributed activity should not convey any information. Yet, the null data show more position information per spike during sparser phases (Fig. 5G). To correct for this bias, we follow previous protocols and subtract averages over many null-matched samples from position information[48]. In all of our comparisons of information between sparse and dense phases, including the model prediction in Fig. 5A, we report sparsity-corrected information. For further validation, we generate a shuffled dataset that disrupts any relationship between spike positions and phases found in the original data (Fig. 5I). Figure 5J, K illustrates a second place field whose tuning also depends on theta phase but does not exhibit precession. For each theta-modulated CA3 place field, we partition phases into sparse and dense halves based on activity, and we average the sparsity-corrected position information per spike across each partition. CA3 place fields convey significantly more information during sparse phases than dense phases (Fig. 5L). This relationship is present in both phase-precessing and other fields (although slightly non-significantly in the latter) and is absent in the shuffled data. Thus, experimental data support our model's prediction that CA3 encodes information in a finer, example-like manner during sparse theta phases. Notably, CA1 place fields do not convey more information per spike during sparse phases, which helps to show that our prediction is nontrivial and demonstrates that the phase behavior in CA3 is not just simply propagated forward to CA1 (Supplementary Fig. 5C).

To characterize the relationship between information and theta phase more precisely, we aggregate spikes over phase-precessing

fields in CA3 and in CA1 (Supplementary Fig. 5D–G). This process implicitly assumes that each phase-precessing field is a sample of a general distribution characteristic to each region. These aggregate fields recapitulate the single-neuron results that CA3 spikes are uniquely more informative during sparse phases (Supplementary Fig. 5H). They also reveal how position information varies with other field properties over theta phases (Supplementary Fig. 5I, J). For example, information is negatively correlated with field width, confirming the interpretation that more informative phases have sharper tuning curves (Fig. 5E). In CA3, information is greatest during early progression through the field, which corresponds to future locations, with a smaller peak during late progression, which corresponds to past locations. In contrast, past locations are more sharply tuned in CA1. Thus, different hippocampal subfields may differentially encode past and future positions across the theta cycle; we will return to this topic in the Discussion.

Next, we turn our attention to the representation of concepts instead of examples. Our model predicts that single neurons exhibit more concept information per spike during dense activity regimes (Fig. 6A). To test this prediction using the same CRCNS hc-3 dataset, we invoke the aforementioned equivalence between concepts in our model and coarser positions along a linear track (Fig. 6B). Thus, single CA3 neurons should encode more information per spike about coarse positions during dense theta phases. Previously, to test for finer position encoding in Fig. 5, we divided single place fields into multiple position bins during the computation of information. Here, we analyze encoding of coarser positions by choosing large position bins across the whole track (Fig. 6C, D). We consider different bin sizes to characterize at which scale the merging of examples into concepts occurs. When we again compute the average difference in sparsity-corrected position information per spike between sparse and dense theta phases, we find that dense phases are the most preferentially informative at the coarsest scales (Fig. 6E). CA1 place cells also exhibit this property (Supplementary Fig. 6B). Crucially, differences between sparse and dense phases are not seen in shuffled data, which supports the validity of our analysis methods (Fig. 6F). Our results are further bolstered by their preservation under a different binning procedure (Supplementary Fig. 6C–E). Thus, coarse positions along a linear track can be best distinguished during dense theta phases, in agreement with our model. Note that we always consider 4 bins at a time even for track scales smaller than 1/4, because changing the number of bins across scales introduces a bias in the shuffled data (Supplementary Fig. 6F–H). At finer scales, this process sometimes fails to capture entire fields and artificially splits them (Fig. 6C, left). Here, we adopt a neutral approach and do not adjust our partitions to avoid these cases; the opposite approach was adopted in Fig. 5 where intact fields were explicitly extracted. These different approaches may explain why at the finest scales in Fig. 6E, sparse phases do not convey more information like they do in Fig. 5L.

In our model, concepts are formed by merging examples across all correlated features. While track position can be one such feature, we now assess whether our predictions also apply to another one. In the CRCNS hc-6 dataset contributed by Loren Frank and colleagues, CA3 place cells are recorded during a W-maze alternation task in which mice must alternately visit left and right arms between runs along the center arm[25]. It is known that place cells along the center arm can encode the turn direction upon entering or leaving the center arm in addition to position[49,50]. Again, our model predicts that sparse theta phases preferentially encode specific information (Fig. 7A), so they should be more tuned to a particular turn direction (Fig. 7B). During dense phases, they should generalize over turn directions and solely encode position.

Figure 7C shows spikes from one CA3 place cell accumulated over outward runs along the center arm followed by either left or right turns (Supplementary Fig. 7A). For each theta phase, we compute the total

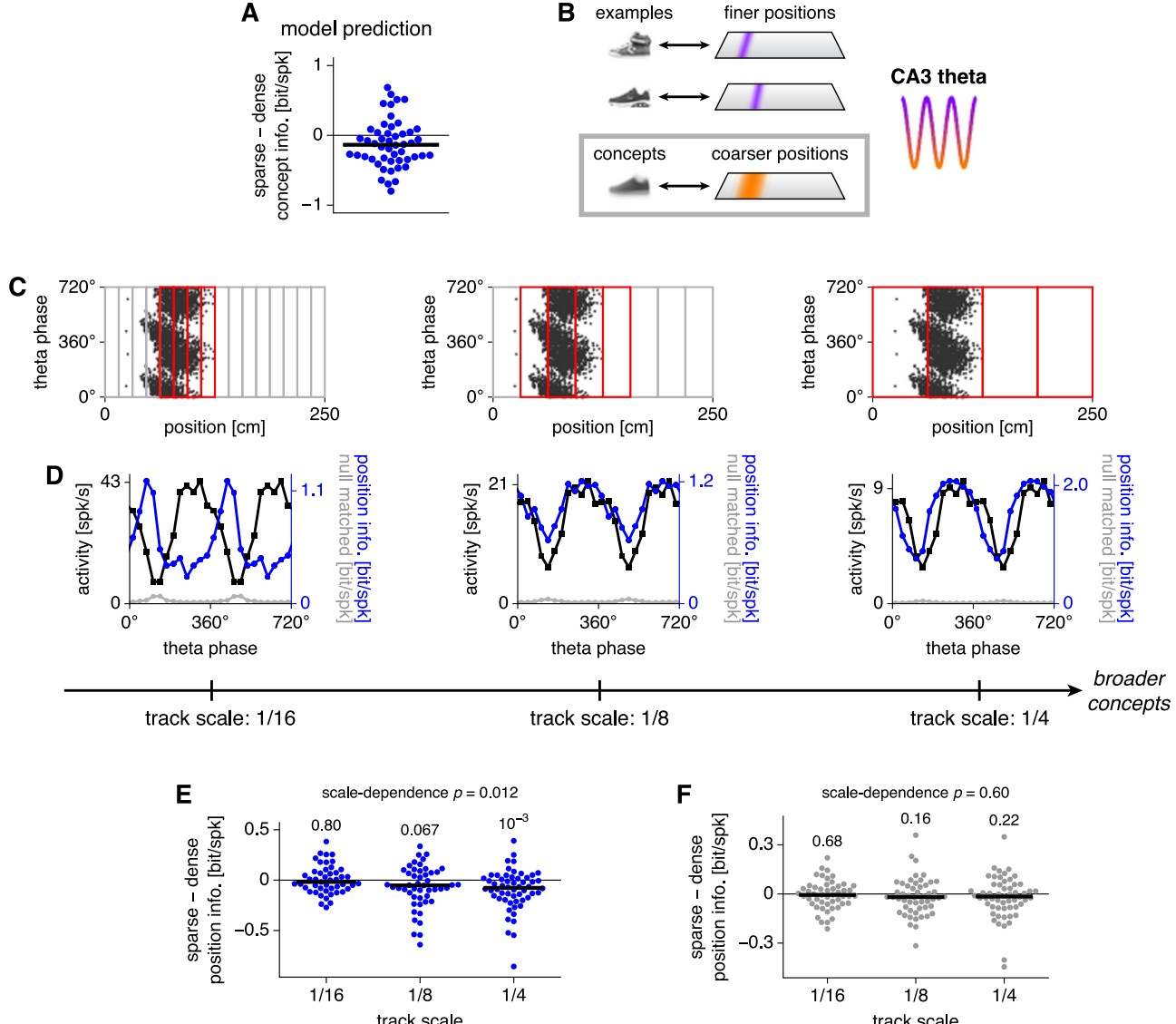

**Fig. 6 | Place cell data support the model prediction that denser theta phases should preferentially encode coarser, concept-like positions. A** Our CA3 model predicts that single neurons convey more information per spike about concept identity during dense regimes. Each point represents a neuron, $n = 50$. **B** To test our model, we construe CA3 place cells to store fine positions as examples, which can combine into coarser regions as concepts. Here, we focus on concept encoding. **C** We calculate position information at various track scales over windows of 4 contiguous bins. **D** Activity (black), raw position information per spike (blue), and mean null-matched position information (gray) by theta phase for the red windows in **C**. Sparsity-corrected position information is the difference between the raw and mean null-matched values. **E** Average difference in position information between the sparsest and densest halves of theta phases. For coarser scales, dense phases convey more position information per spike. Each point represents a place cell averaged over all windows. Track scale 1/16 $n = 47$, 1/8 $n = 49$, and 1/4 $n = 56$. Numbers indicate $p$-values calculated by two-tailed Wilcoxon signed-rank tests for each scale and by Spearman's $\rho$ for the trend across scales. **F** Similar to **E**, but for shuffled data whose spike phases and positions are permuted. For all results, spikes during each traveling direction are separately analyzed. In **A**, **E**, and **F**, information is sparsity-corrected with horizontal lines indicating medians. Source data are provided as a Source Data file.

activity, the turn information per spike (ignoring position), and the mean information of null-matched samples used for sparsity correction (Fig. 7D). Figure 7E, F shows similar results for inward runs (for others, see Supplementary Fig. 7B, C). For both outward and inward runs, sparsity-corrected turn information per spike is greater during sparse theta phases compared to dense phases (Fig. 7G). This finding is not observed in data in which theta phase and turn direction are shuffled (Fig. 7G, H). Not only do these results support our model, they also reveal that in addition to splitter cells that encode turn direction over all theta phases[51], CA3 contains many more place cells that encode it only at certain phases (Supplementary Fig. 7D). The difference between sparse and dense phases is significantly greater in CA3 than it is in CA1 (Supplementary Fig. 7E, F). Thus, our subfield-specific results

for example encoding are consistent across position and turn direction. Aggregate neurons, formed by combining spikes from more active turn directions and those from less active turn directions, demonstrate similar tuning properties to individual neurons (Supplementary Fig. 7G–I).

Beyond the single-neuron results presented above, we seek to test our predictions at the population level. To do so, we perform phase-dependent Bayesian population decoding of turn direction during runs along the center arm (Fig. 7I). This analysis requires multiple neurons with sufficiently sharp tuning to be simultaneously active across all theta phases; it can be used to decode left versus right turns, whereas an analogous decoding of track position, which spans a much broader range of values, is intractable with our datasets. We find that

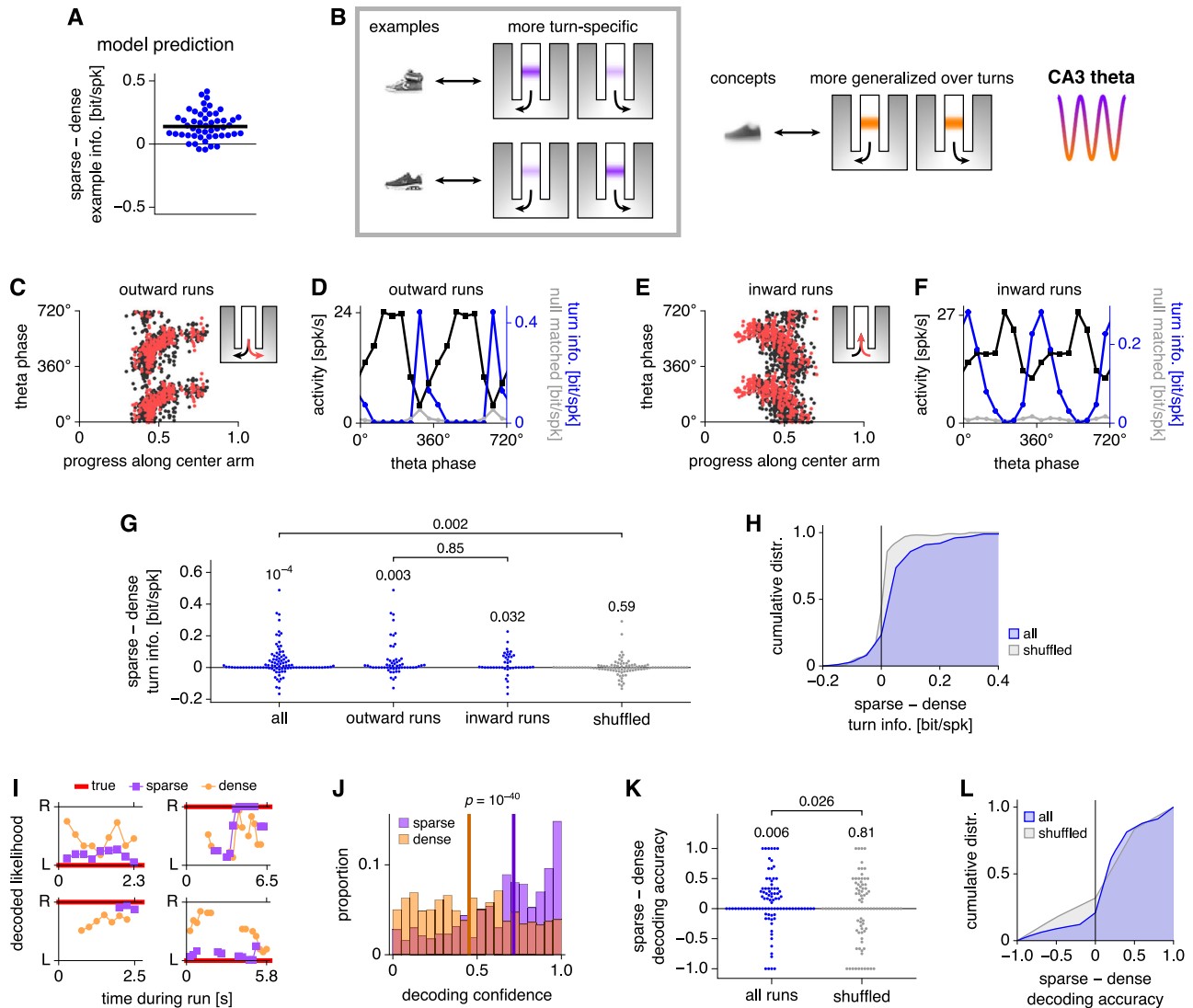

**Fig. 7 | W-maze data support the model prediction that sparser theta phases should preferentially encode turn direction in addition to position. A** Same as Fig. 5A. **B** To test our model, we construe CA3 place cells to store turn directions during the central arm of a W-maze alternation task as examples. By combining examples, concepts that generalize over turns to solely encode position can be formed. **C–H** Single-neuron information results. **C** Example place cell that is active during outward runs. Each spike is represented by two points at equivalent phases with different colors representing different future turn directions. **D** Activity (black), raw turn information (blue), and mean null-matched turn information (gray) by theta phase for the neuron in **C**. Sparsity-corrected turn information is the difference between the raw and mean null-matched values. **E, F** Similar to **C, D**, but for a place cell that is active during inward runs with colors representing past turn directions. **G** Average difference in turn information between the sparsest and densest halves of theta phases. For all cell populations, sparse phases convey more turn information per spike. Each point represents a place cell. All and shuffled n = 99, outward runs n = 56, and inward runs n = 43. Numbers indicate p-values

calculated by two-tailed Wilcoxon signed-rank tests except for the comparison between outward and inward runs, which is calculated by the two-tailed Mann-Whitney U test. **H** Cumulative distribution functions for values in **G**. **I–L** Bayesian population decoding results. **I** Likelihood of left (L) or right (R) turns during four runs along the center arm using spikes from either the sparsest or densest halves of theta phases. **J** Sparse encodings exhibit greater confidence about turn direction. Vertical lines indicate medians with p-value calculated by the two-tailed Mann-Whitney U test. **K** Average difference in maximum likelihood estimation accuracy between the sparsest and densest halves of theta phases. Sparse phases encode turn direction more accurately. Each point represents one run averaged over decoded timepoints. All runs and shuffled n = 91. Numbers indicate p-values calculated by two-tailed Wilcoxon signed-rank tests. **L** Cumulative distribution functions for values in **K**. For all results, spikes during each traveling direction are separately analyzed. In **A, G**, and **H**, information is sparsity-corrected. Source data are provided as a Source Data file.

the CA3 population likelihood exhibits greater confidence during sparse phases (Fig. 7J). From a Bayesian perspective, the population expresses stronger beliefs about turn direction during sparse phases and is more agnostic during dense phases. If pressed to choose the direction with a higher likelihood as its estimate, CA3 is also more accurate during sparse phases (Fig. 7K, L). These results match our predictions in Fig. 7A, B and bolster our single-neuron results.

Moreover, they are specific to CA3, as similar conclusions cannot be made about the CA1 place cell population (Supplementary Fig. 7J–L).

In summary, extensive data analysis reveals experimental support for our CA3 model over two datasets collected by different research groups, across two encoding modalities, for both example and concept representations, and at both the single-neuron and population level.

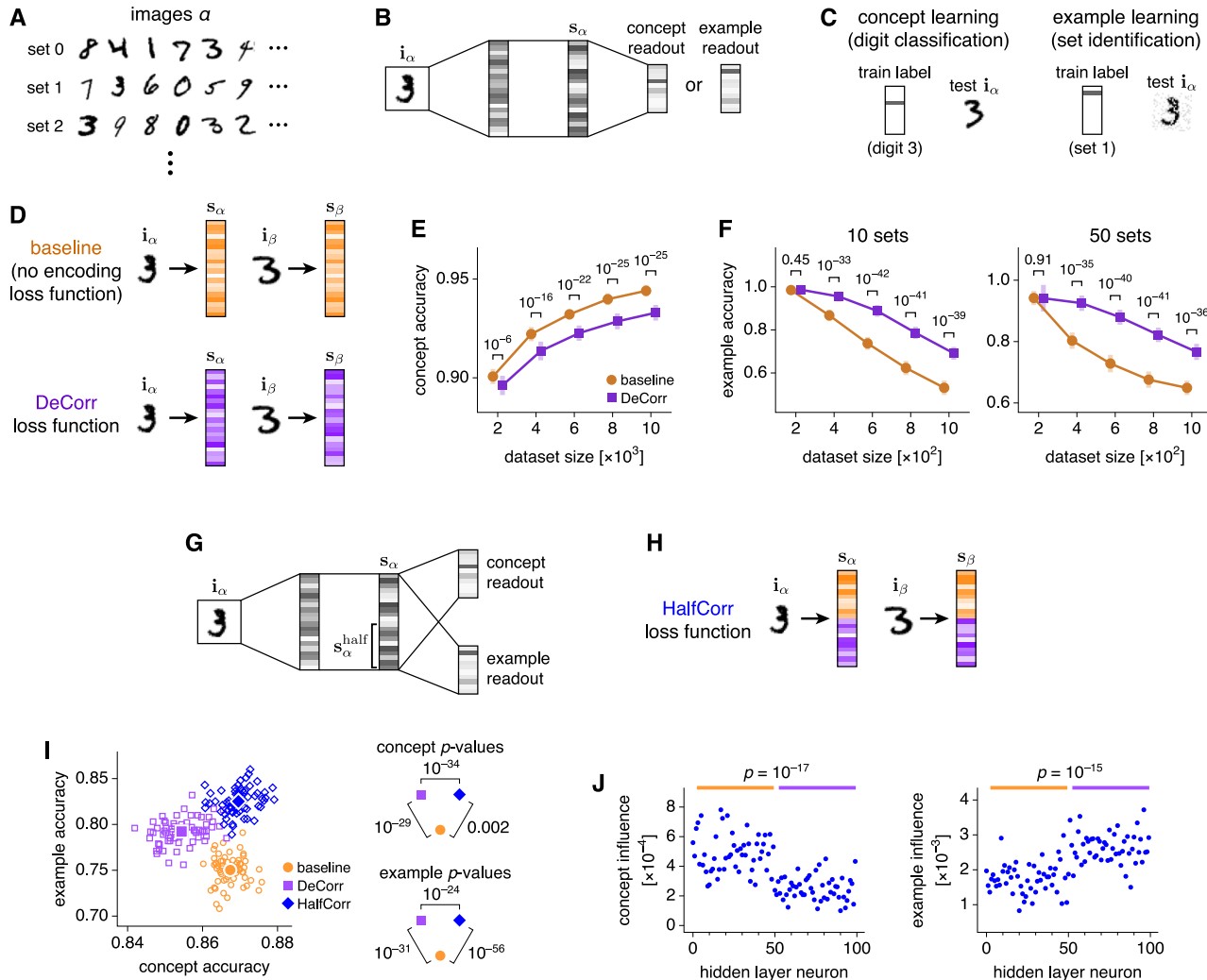

**Fig. 8 | Complementary encodings inspired by CA3 can improve machine learning performance in a complex task. A** We extend the MNIST dataset by randomly assigning an additional set label to each image. **B–F** We train a multilayer perceptron to either classify digits or identify sets. **B** Network architecture. Each hidden layer contains 50 neurons. **C** Task structures. Digit classification requires building concepts and is tested with held-out test images. Set identification requires distinguishing examples and is tested with noisy train images. **D** We apply the DeCorr loss function (Eq. (3)) to decorrelate encodings in the final hidden layer, in analogy with MF patterns in CA3. Without an encoding loss function, image correlations are preserved, in analogy with PP patterns. **E, F** DeCorr decreases concept learning performance and increases example learning performance. Points indicate means and bars indicate standard deviations over 32 networks. **G–J** We train a multilayer perceptron to simultaneously classify digits and identify sets. **G** Network architecture. Each hidden layer contains 100 neurons. The train dataset contains 1000 images and 10 sets. **H** We apply the HalfCorr loss function (Eq. (4)) to decorrelate encodings only among the second half of the final hidden layer. Correlated and decorrelated encodings are both present, in analogy with MF and PP patterns across the theta cycle in CA3. **I** DeCorr networks generally perform better at example learning but worse at concept learning compared to baseline. HalfCorr networks exhibit high performance in both tasks. Open symbols represent individual networks and filled symbols represent means over 64 networks. **J** Influence of each neuron in HalfCorr networks on concept and example learning, defined as the average decrease in accuracy upon clamping its activation to 0. Correlated neurons (orange bars) are more influential in concept learning, and decorrelated neurons (purple bars) are more influential in example learning. For all results, $p$-values are computed using unpaired two-tailed $t$-tests. Source data are provided as a Source Data file.

## CA3-like complementary encodings improve neural network performance in multitask machine learning

We have observed how CA3 encodes behaviorally relevant information at different scales across theta phases. Can these different types of encodings be useful for solving different types of tasks? Can they even benefit neural networks designed for machine learning, abstracting away from the hippocampus? To address these questions, we turn to a classic paradigm in machine learning: a multilayer perceptron trained on MNIST handwritten digit images[52]. First, we augment the MNIST dataset by randomly assigning an additional label to each image: a set number (Fig. 8A). We train the fully connected feedforward network to perform one of two tasks: classification of the written digit or identification of the assigned set (Fig. 8B). The former requires clustering of

images based on common features, which resembles concept learning in our CA3 model, and the latter requires discerning differences between similar images, which resembles example learning in our CA3 model (Fig. 8C). We use a held-out test dataset to evaluate digit classification performance and corrupted images from the train dataset to evaluate set identification performance.

In our CA3 model, we found that examples were preferentially encoded by the decorrelated MF pathway and concepts by the correlated PP pathway (Fig. 3). In an analogous fashion, we seek to manipulate the correlation properties within the final hidden layer of our perception, whose activations $s_\alpha$ serve as encodings of the input images $i_\alpha$. In particular, we apply a DeCorr loss function, which penalizes correlations in $s_\alpha$ between every pair of items $\alpha$, $\beta$ in a training

batch (Fig. 8D):

$$\mathcal{L}_{\mathrm{DeCorr}} \approx \frac{1}{2} \sum_{\substack{\alpha,\beta \in \\ \mathrm{batch}}} \mathrm{Pearson}(\mathbf{s}_\alpha, \mathbf{s}_\beta)^2. \tag{3}$$

DeCorr mimics the MF pathway; the equation is approximate due to a slight modification of the Pearson correlation formula to aid numerical convergence (Methods). Alternatively, we consider the baseline condition with no loss function on hidden layer activations, which preserves natural correlations between similar images and mimics the PP pathway. Indeed, we observe that different encoding properties are suited for different tasks. Baseline networks perform better in concept learning (Fig. 8E) while DeCorr networks perform better in example learning (Fig. 8F), and these effects vary consistently with the strength of the DeCorr loss function (Supplementary Fig. 8A, B). Thus, DeCorr allows us to tune encoding correlations in neural networks to highlight input features at either broader or finer scales. Tasks can be solved more effectively by matching their computational requirements with the appropriate encoding scale. Note that DeCorr is different from the DeCov loss function previously developed to reduce overfitting[53]. DeCorr decorrelates pairs of inputs across all neurons in the specified layer, whereas DeCov decorrelates pairs of neurons across all inputs. As a regularizer that promotes generalization, DeCov improves digit classification and does not substantially improve set identification, which contrasts with the effect of DeCorr (Supplementary Fig. 8C, D).

Complex tasks, including those performed by biological systems, may require information to be processed at different scales of correlation. In CA3, a spectrum of encodings is available during each theta cycle. Can neural networks take advantage of multiple encodings? We tackle this question by asking a perceptron to simultaneously perform digit classification and set identification (Fig. 8G). In addition to the baseline and DeCorr networks, we define a HalfCorr loss function (Fig. 8H):

$$\mathcal{L}_{\mathrm{HalfCorr}} \approx \frac{1}{2} \sum_{\substack{\alpha,\beta \in \\ \mathrm{batch}}} \mathrm{Pearson}(\mathbf{s}_\alpha^{\mathrm{half}}, \mathbf{s}_\beta^{\mathrm{half}})^2, \tag{4}$$

where $\mathbf{s}_\alpha^{\mathrm{half}}$ represents the second half of neurons in the final hidden layer. After training with this loss function, the neural representation consists of both a correlated, PP-like component in the first half and a decorrelated, MF-like component in the second half. When we evaluate these networks on both digit classification and set identification, we see that baseline and DeCorr networks behave similarly to how they did on single tasks. Compared to baseline, DeCorr networks perform better in example learning at the cost of poorer concept learning (Fig. 8I). However, HalfCorr networks do not suffer from this tradeoff and perform well at both tasks. Their superior performance is maintained over a variety of network and dataset parameters (Supplementary Fig. 8E). Moreover, HalfCorr networks learn to preferentially use each type of encoding for the task to which it is better suited. We use the decrease in task accuracy upon silencing a neuron as a metric for its influence on the task. Correlated neurons are more influential in concept learning and decorrelated neurons in example learning (Fig. 8J).

Note that we do not manipulate pattern sparsity in these artificial networks. Sparsification can be useful in the hippocampus because it provides a biologically tractable means of achieving decorrelation. It also allows biological networks to access both less and more correlated representations by changing the level of inhibition. Instead, we can directly manipulate correlation through the DeCorr and HalfCorr loss functions. Under some conditions, the decorrelated half of the final hidden layer in HalfCorr networks indeed exhibits sparser activation than the correlated half (Supplementary Fig. 8F). It is possible that directly diversifying sparsity can also improve machine learning

performance, especially since sparse coding is known to offer certain computational advantages as well as greater energy efficiency[27,54,55].

## Discussion

The hippocampus is widely known to produce our ability to recall specific vignettes as episodic memories. This process has been described as indexing every sensory experience with a unique neural barcode so that separate memories can be independently recovered[56,57]. Recently, research has shown that the hippocampus is also important in perceiving commonalities and regularities across individual experiences, which contribute to cognitive functions such as statistical learning[58,59], category learning[60–63], and semantic memory[64–66]. Evidence for this has been obtained largely through human studies, which can present and probe memories in controlled settings. However, the detailed circuit mechanisms used by the hippocampus to generalize across experiences while also indexing them separately are not known.

Our analysis of rodent place cell recordings reveals that single CA3 neurons alternate between finer, example-like representations and broader, concept-like representations of space across the theta cycle (Figs. 5–7). These single-neuron results extend to the network level, which alternatively encodes more specific and more general spatial features in a corresponding manner (Fig. 7). If we accept that place cells store these features as spatial memories, then our experimental analysis reveals that CA3 can access memories of different scales at different theta phases. We propose that the computational mechanism underlying these observations is the multiplexed encoding of each memory at different levels of correlation (Figs. 2 and 8). We show that CA3 can biologically implement this mechanism through the storage of both sparse, decorrelated MF inputs and dense, correlated PP inputs and their alternating retrieval by the theta oscillation, which acts as an activity threshold and subtractively modulates neural activity (Fig. 2, 3, and 4). Our model performs successful pattern completion of both types of encodings, suggesting that patterns across the theta cycle can truly function as memories that can be recovered from partial cues.

Alone, our secondary analyses of experimental data contribute to a large set of observations on how coding properties vary with theta phase in the hippocampus. Of note is phase precession, in which different phases preferentially encode different segments within a firing field as it is traversed, with later phases tuned to earlier segments[67]. Phase precession is most widely reported for place cells and traversals of physical space, but it also appears during the experience of other sequences, such as images and tasks[68–70]. Our analysis implies that the sharpness of tuning is not constant throughout traversals. In particular, CA3 neurons are more sharply tuned at early positions in place fields, while CA1 neurons are more sharply tuned at late positions (Supplementary Fig. 5I, J). Transforming these conclusions about position into those about time through the concept of theta sequences, CA3 represents the future more precisely, while CA1 represents the future more broadly. The latter is consistent with the idea that CA1 may participate in the exploration of multiple possible future scenarios[71]. Furthermore, our W-maze analysis reveals that certain hippocampal neurons which do not obviously encode an external modality across all theta phases, such as turn direction, may do so only during sparse phases (Supplementary Fig. 7D). This observation adds to the subtleties with which the hippocampus represents the external world.

Other groups have investigated the variation of place field sharpness with theta phase in CA1, not CA3, and their results are largely in agreement with our CA1 analyses. Skaggs et al.[43] partitioned theta phases into halves, one of which with higher activity than the other, and found more information per spike during the less active half. We observe no difference at the single-neuron level, though our W-maze results are only slightly non-significant (Supplementary Figs. 5C and 7E). Their partitions differ from ours by 30° and they employ a different

binning technique, both of which can influence the results. The less informative phases in their work correspond to future positions, which we also observe (Supplementary Fig. 5J). Note that their computation of sparsity is performed along a different axis compared to ours; using terms from Willmore and Tolhurst[72], they use the *lifetime* density while we compute the *population* density, which fundamentally differ. Ujfalussy and Orbán[73] also found that phases with larger field sizes correspond to future positions. Mehta et al.[74] considered phase-dependent tuning within CA1 place fields, but they calculate field width over theta phase as a function of field progress, whereas we do the opposite. Similarly, Souza and Tort[75] considered phase tuning at various field progresses. Both sets of results appear to be compatible with ours, but a direct comparison cannot be made. Overall, our work offers original insights into hippocampal phase coding not only by focusing on CA3, which behaves differently from CA1, but also by elucidating a connection between tuning width and network sparsity. Intriguingly, Pfeiffer and Foster[76] found a relationship between CA1 replay speed and slow gamma oscillation phase, which modulates network activity during quiescence. This observation opens the possibility that other oscillations can leverage the connection between tuning and sparsity when the dominant theta rhythm is absent.

Better memory performance has been associated with greater theta power during both encoding[77–82] and retrieval[79,82–84]. Our model cannot explain the former because it does not contain a theta oscillation during encoding. It does, however, offer a possible explanation for the latter observation. We simulated retrieval under an oscillating threshold with lower amplitude and observed that the network stalls on MF example encodings instead of alternating with PP concept encodings (Fig. 4A). Thus, the biological processes that produce larger theta amplitudes, such as stronger medial septum inputs or changes in neuromodulator concentrations[85], may promote memory recall by granting access to wider ranges of pattern sparsities and, consequently, representational scales.

The temporal coordination between memory storage and retrieval is also biologically significant. We make the major simplification of separately simulating memory storage and retrieval. These two operating regimes can represent different tones of a neuromodulator such as acetylcholine, which is thought to bias the network towards storage[86]. Another proposal is that storage and retrieval preferentially occur at different theta phases, motivated by the variation in long-term potentiation (LTP) strength at CA1 synapses across the theta cycle[87–89]. Although this idea focuses on plasticity in CA1, it is possible that storage and retrieval also occur at different phases in CA3. Note that our experimental analysis reveals a sharp dip in position information around the sparsest theta phase in both CA3 and CA1 (Supplementary Fig. 5E, G). This phase may coincide with the storage of new inputs, during which the representation of existing memories is momentarily disrupted; the rest of the theta cycle may correspond to retrieval. This interpretation could motivate excluding the sparsest theta phase from further analysis, since our model predictions only regard memory retrieval. However, we take a conservative approach and include all phases. Interestingly, recent work reported that the strength of LTP in CA1 peaks twice per theta cycle[90], suggesting for our model that MF and PP patterns could have their own storage and retrieval intervals during each theta cycle.

Our work connects hippocampal anatomy and physiology with foundational attractor theory. Among others, Tsodyks and Feigel'man[91] observed that sparse, decorrelated patterns can be stored at high capacity, and Fontanari[92] found that dense, correlated patterns can merge into representations of common features. We demonstrate that both types of representations can be stored and retrieved in the same network, using a threshold to select between them. This capability can be given solid theoretical underpinnings using techniques from statistical mechanics[41]. The convergence of MF and PP pathways in CA3 has also been the subject of previous computational

investigations[15–17]. In these models, CA3 stores and retrieves one encoding per memory, while our model asserts that multiple encodings for the same memory alternate across the theta cycle. Another series of models proposes, like we do, that the hippocampus can simultaneously maintain both decorrelated, example-like encodings and correlated, concept-like encodings[93,94]. These encodings converge at CA1 and each type is not independently retrieved there, which differs from our model. In a related hippocampal model, the PP pathway was shown to be crucial for learning cue-target associations in the presence of additional context inputs to EC that drift over time[95]. Successful learning requires the network to perform decontextualization and abstract away the slowly varying context inputs, which is, like concept learning, a form of generalization. EC has also been hypothesized to differentially encode inputs upstream of CA3, with specific sensory information conveyed by lateral EC and common structural representations by medial EC[96]. Further experimental investigation into the contributions of various subregions would help to clarify how the hippocampus participates in memory generalization.

Finally, our work addresses how CA3-like complementary encodings can be computationally leveraged by neural networks to solve complex tasks. We conceptually extend our results about CA3 and introduce a HalfCorr loss function that diversifies hidden layer representations to include both correlated and decorrelated components (Fig. 8). HalfCorr networks can better learn tasks that involve both distinction between similar inputs and generalization across them. They are simultaneously capable of pattern separation and categorization even based on small datasets, demonstrating a possible advantage of brain computation over conventional deep learning. Yet, we deliberately chose a neural architecture that differs from that of the CA3 network to test the scope over which complementary encodings can improve learning. Instead of a recurrent neural network storing patterns of different sparsities through unsupervised Hopfield learning rules, we implemented a feedforward multilayer perceptron, a workhorse of supervised machine learning. The success of HalfCorr networks in this scenario supports the possibility that HalfCorr can be broadly applied as a plug-and-play loss function to improve computational flexibility.

Functional heterogeneity is commonly invoked in the design of modern neural networks. It can be implemented in the form of deep or modular neural networks in which different subnetworks perform different computations in series or parallel, respectively[97,98]. Of note, Kowadlo et al.[99] constructed a deep, modular network inspired by the hippocampus for one-shot machine learning of both concepts and examples. As an aside, their architecture shares similarities with our hippocampus model in Fig. 2, but their Hopfield-like network for CA3 only stores MF patterns and inactivating these recurrent connections does not affect network performance. In contrast to these networks with specialized subnetworks, we propose a different paradigm in which a loss function applied differentially across neurons promotes heterogeneity within a single layer. This idea can be extended from the two components of HalfCorr networks, correlated and decorrelated, by assigning a different decorrelation strength to each neuron and thereby producing a true spectrum of representations. Furthermore, heterogeneity in other encoding properties such as mean activation, variance, and sparsity may also improve performance in tasks with varying or unclear computational requirements. Such tasks are not limited to multitask learning, but also include continual learning[100], reinforcement learning[101], and natural learning by biological brains.

## Methods

### Transformation of memories along hippocampal pathways

**Binary autoencoder from images to EC.** Our memories are 256 images from each of the *sneaker*, *trouser*, and *coat* classes in the FashionMNIST dataset[26]. We train a fully connected linear autoencoder on these images with three hidden layers of sizes 128, 1024, and 128. Batch

normalization is applied to each hidden layer, followed by a rectified linear unit (ReLU) nonlinearity for the first and third hidden layers and a sigmoid nonlinearity for the output layer. Activations in the middle hidden layer are binarized by a Heaviside step function with gradients backpropagated by the straight-through estimator[102]. The loss function is

$$\mathcal{L} = \sum_{\substack{\mu\nu \in \\ \text{batch}}} ||\hat{\mathbf{i}}_{\mu\nu} - \mathbf{i}_{\mu\nu}||^2 + \lambda \sum_{\substack{\mu\nu \in \\ \text{batch}}} \text{KL}\left(\frac{1}{N_{\text{EC}}} \sum_i x^{\text{EC}}_{\mu\nu i} \,\Big|\Big|\, a_{\text{EC}}\right), \tag{5}$$

where $\mathbf{i}_{\mu\nu}$ is the image with pixel values between 0 and 1, $\hat{\mathbf{i}}_{\mu\nu}$ is its reconstruction, $\mathbf{x}^{\text{EC}}_{\mu\nu}$ represents the binary activations of the middle hidden layer with $N_{\text{EC}} = 1024$ units indexed by $i$, and $a_{\text{EC}} = 0.1$ is its desired density (Fig. 2C). Sparsification with strength $\lambda = 10$ is achieved by computing the Kullback-Leibler (KL) divergence between the hidden layer density and $a_{\text{EC}}$[103]. Training is performed over 150 epochs with batch size 64 using the Adam optimizer with learning rate $10^{-3}$ and weight decay $10^{-5}$.

**Binary feedforward networks from EC to CA3.** To propagate patterns from EC to DG, from DG to MF inputs, and from EC to PP inputs, we compute

$$x^{\text{post}}_{\mu\nu i} = \Theta\left[\sum_j W_{ij} x^{\text{pre}}_{\mu\nu j} - \theta\right], \tag{6}$$

where $\mathbf{x}^{\text{pre}}_{\mu\nu}$ and $\mathbf{x}^{\text{post}}_{\mu\nu}$ are presynaptic and postsynaptic patterns, $W_{ij}$ is the connectivity matrix, and $\theta$ is a threshold. Each postsynaptic neuron receives $l$ excitatory synapses of equal strength from randomly chosen presynaptic neurons. $\theta$ is implicitly set through a winners-take-all process that enforces a desired postsynaptic pattern density $a_{\text{post}}$. $\Theta$ is the Heaviside step function, and $N$ is the network size.

EC patterns have $N_{\text{EC}} = 1024$ and $a_{\text{EC}} = 0.1$. To determine $N$, $a$, and $l$ for each subsequent region, we turn to estimated biological values and loosely follow their trends. Rodents have approximately 5–10 times more DG granule cells and 2–3 times more CA3 pyramidal cells compared to medial EC layer II principal neurons[14,104,105]. Thus, we choose $N_{\text{DG}} = 8192$ and $N_{\text{CA3}} = 2048$. During locomotion, DG place cells are approximately 10 times less active than medial EC grid cells[106], and MF inputs are expected to be much sparser than PP inputs[15]. Thus, we choose $a_{\text{DG}} = 0.005$, $a_{\text{MF}} = 0.02$, and $a_{\text{PP}} = 0.2$. We do not directly enforce correlation within concepts, which take values $\rho_{\text{EC}} = 0.15$, $\rho_{\text{DG}} = 0.02$, $\rho_{\text{MF}} = 0.01$, and $\rho_{\text{PP}} = 0.09$. Each DG neuron receives approximately 4000 synapses from EC and each CA3 neuron receives approximately 50 MF and 4000 PP synapses[14]. Thus, we choose $l_{\text{DG}} = 205$, $l_{\text{MF}} = 8$, and $l_{\text{PP}} = 205$. Note that for each feedforward projection, postsynaptic statistics $a_{\text{post}}$ and $\rho_{\text{post}}$ are not expected to depend on $l$ (Eq. (1)).

In Fig. 2E, for the case of $\mathbf{x}^{\text{pre}}_{\mu\nu} = \mathbf{x}^{\text{EC}}_{\mu\nu}$, we use $N_{\text{post}} = 2048$ and $l = 205$. $\rho_{\text{post}}$ is obtained by computing correlations between examples within the same concept and averaging over 3 concepts and 8 connectivity matrices. For the case of randomly generated $\mathbf{x}^{\text{pre}}_{\mu\nu}$, we use $N_{\text{pre}} = N_{\text{post}} = 10000$, a single concept, and a single connectivity matrix with $l = 2000$. See Supplementary Information for further details, including the derivation of Eq. (1).

**Visualization pathway from CA3 to EC.** We train a fully connected linear feedforward network with one hidden layer of size 4096 to map inputs $\mathbf{x}^{\text{MF}}_{\mu\nu}$ to targets $\mathbf{x}^{\text{EC}}_{\mu\nu}$ and inputs $\mathbf{x}^{\text{PP}}_{\mu\nu}$ also to targets $\mathbf{x}^{\text{EC}}_{\mu\nu}$. Batch normalization and a ReLU nonlinearity is applied to the hidden layer and a sigmoid nonlinearity is applied to the output layer. The loss

function is

$$\mathcal{L} = \sum_{\substack{\mu\nu \in \\ \text{batch}}} ||\hat{\mathbf{x}}^{\text{EC}}_{\mu\nu} - \mathbf{x}^{\text{EC}}_{\mu\nu}||^2. \tag{7}$$

Training is performed over 100 epochs with batch size 128 using the Adam optimizer with learning rate $10^{-4}$ and weight decay $10^{-5}$.

**Hopfield-like model for CA3**

**Pattern storage.** Our Hopfield-like model for CA3 stores linear combinations $\mathbf{q}_{\mu\nu}$ of MF and PP patterns:

$$q_{\mu\nu i} = (1 - \zeta) \cdot (x^{\text{MF}}_{\mu\nu i} - a_{\text{MF}}) + \zeta \cdot (x^{\text{PP}}_{\mu\nu i} - a_{\text{PP}}), \tag{8}$$

where $\zeta = 0.1$ is the relative strength of the PP patterns (Fig. 3A). The subtraction of densities from each pattern is typical of Hopfield networks with neural states 0 and 1[91]. The synaptic connectivity matrix is

$$W_{ij} = \frac{1}{N_{\text{CA3}}} \sum_{\mu\nu} q_{\mu\nu i} q_{\mu\nu j}. \tag{9}$$

**Pattern retrieval.** Cues are formed from target patterns by randomly flipping the activity of a fraction 0.01 of all neurons (Fig. 3B). This quantity is termed *cue inaccuracy*; in Supplementary Fig. 3F, we also consider *cue incompleteness*, which is the fraction of active neurons that are randomly inactivated to form the cue. During retrieval, neurons are asynchronously updated in cycles during which every neuron is updated once in random order (Fig. 3C). The total synaptic input to neuron $i$ at time $t$ is

$$g_i(t) = \sum_j W_{ij} S_j(t) + h_i(t), \tag{10}$$

where $S_j(t)$ is the activity of presynaptic neuron $j$ and $h_i(t)$ is an external input. The external input is zero except for the cue-throughout condition in Fig. 4, in which $\mathbf{h}(t) = \sigma\mathbf{x}$ for noisy MF cue $\mathbf{x}$ and strength $\sigma = 0.2$.

The activity of neuron $i$ at time $t$ is probabilistically updated via the Glauber dynamics

$$P[S_i(t+1) = 1] = \frac{1}{1 + e^{-\beta[g_i(t) - \theta(t)]}}, \tag{11}$$

where $\theta$ is the threshold and $\beta$ is inverse temperature, with higher $\beta$ implying a harder threshold. Motivated by theoretical arguments, we define rescaled variables $\theta'$ and $\beta'$ such that $\theta = \theta' \cdot (1 - \zeta)^2 a_{\text{MF}}$ and $\beta = \beta'/(1 - \zeta)^2 a_{\text{MF}}$[41]. Unless otherwise indicated, we run simulations for 10 update cycles, use $\beta' = 100$, and use $\theta' = 0.5$ to retrieve MF patterns and $\theta' = 0$ to retrieve PP patterns. The rescaled $\theta'$ is the threshold value illustrated in Fig. 4A and Supplementary Fig. 4A.

**Retrieval evaluation.** The overlap between the network activity $\mathbf{S}$ and a target pattern $\mathbf{x}$ is

$$m = \frac{1}{N_{\text{CA3}} a(1 - a)} \sum_i S_i(x_i - a), \tag{12}$$

where $a$ is the density of the target pattern. This definition is also motivated by theory[41]. The target pattern $\mathbf{x}^{\text{PP}}_\mu$ for PP concept $\mu$ is

$$x^{\text{PP}}_{\mu i} = \Theta\left[\sum_\nu x^{\text{PP}}_{\mu\nu i} - \phi\right], \tag{13}$$

where $\phi$ is a threshold implicitly set by using winners-take-all to enforce that $\mathbf{x}^{\text{PP}}_\mu$ has density $a_{\text{PP}}$. The theoretical maximum overlap

between the network and $\mathbf{x}_\mu^{PP}$ is the square root of the correlation $\sqrt{\rho_{PP}}$[41]. Because this estimate is derived for random binary patterns in the large network limit, it can be exceeded in our simulations.

To visualize $\mathbf{S}$, we first recall that the inverse of a dense stored pattern $\mathbf{x}$ with every neuron flipped can also be an equivalent stable state[33]. Thus, we invert $\mathbf{S}$ if we are retrieving PP patterns at low $\theta$ and if $m < 0$. Then, we decode its EC representation by passing $\mathbf{S}$ through the feedforward visualization network and binarizing the output with threshold 0.5. Finally, we pass the EC representation through the decoding layers of the image autoencoder.

See Supplementary Information for the determination of network capacity with random binary patterns (Fig. 3H, I and Supplementary Fig. 3G, H) and the definition of oscillation behaviors (Fig. 4C and Supplementary Fig. 4B).

### Experimental data analysis

**General considerations.** To calculate activity, we tabulate spike counts $c(r, \phi)$ over spatial bins $r$ (position or turn direction) and theta phase bins $\phi$, and we tabulate trajectory occupancy $u(r, \phi)$ over the same $r$ and distribute them evenly across $\phi$. Activity as a function of theta phase, the spatial variable, and both variables are respectively

$$f(\phi) = \frac{\sum_r c(r,\phi)}{\sum_r u(r,\phi)}, \quad f(r) = \frac{\sum_\phi c(r,\phi)}{\sum_\phi u(r,\phi)}, \quad \text{and} \quad f(r,\phi) = \frac{c(r,\phi)}{u(r,\phi)}. \quad (14)$$

Information per spike as a function of theta phase is calculated by

$$I(\phi) = \sum_r \frac{c(r,\phi)}{c(\phi)} \log_2 \frac{f(r,\phi)}{f(\phi)}, \quad (15)$$

where $c(\phi) = \sum_r c(r, \phi)$[107]. To perform sparsity correction for each neuron, we generate 100 null-matched neurons in which the spatial bin of each spike is replaced by a random value uniformly distributed across spatial bins. We subtract the mean $I(\phi)$ over the null matches from the $I(\phi)$ for the true data. To calculate the average difference in information between sparse and dense phases, we first use $f(\phi)$ to partition $\phi$ into sparse and dense halves. We then average the sparsity-corrected $I(\phi)$ over each half, apply a ReLU function to each half to prevent negative information values, and compute the difference between halves.

See Supplementary Information for dataset preprocessing details.

**Model prediction.** For the example prediction in Fig. 5A, we choose one concept from Fig. 2A and find 50 neurons that are active in at least one MF example and one PP example within it. For each neuron, we convert each active response to one spike and assign equal occupancies across all examples. We calculate the information per spike across MF examples and across PP examples using example identity $v$ as the spatial bin $r$. These values are sparsity-corrected with 50 null-matched neurons, and their difference becomes our example prediction, associating MF encodings with sparse phases and PP with dense.

For the concept prediction in Fig. 6A, we find 50 neurons that are active in at least one MF example and one PP example within any concept. For each neuron, we convert each active response to one spike and collect MF and PP concept responses by summing spikes within each concept. We assign equal occupancies across all concepts. We calculate the information per spike across MF concepts and across PP concepts using concept identity $\mu$ as the spatial bin $r$. We then proceed as in the example case to produce our concept prediction.

**Linear track data.** Single neurons in Fig. 6 are preprocessed from the CRCNS hc-3 dataset as described in Supplementary Information[24]. To identify place cells, we compute the phase-independent position information per spike using 1 cm-bins across all theta phases, and we select neurons with values greater than 0.5. For each place cell, we bin

spikes into various position bins as illustrated and phase bins of width 30°. Since our prediction compares sparse and dense information conveyed by the same neurons, we require at least 8 spikes within each phase value to allow for accurate estimation of position information across all theta phases. To ensure theta modulation, we also require the most active phase to contain at least twice the number of spikes as the least active phase.

Place fields in Fig. 5 are extracted as described in Supplementary Information. Processing occurs similarly to the whole-track case above, except we do not enforce a phase-independent information constraint, we use 5 progress bins, and we require at least 5 spikes within each phase value. Phase precession is detected by performing circular–linear regression between spike progresses and phases[108,109]. The precession score and precession slope are respectively defined to be the mean resultant length and regression slope. Precessing neurons have score greater than 0.3 and negative slope steeper than −72°/field.

**W-maze data.** Single neurons in Fig. 7A–H are preprocessed from the CRCNS hc-6 dataset as described in Supplementary Information[25]. For each neuron, we bin spikes into 2 turn directions and phase bins of width 45°. Since our prediction compares sparse and dense information conveyed by the same neurons, we require at least 5 spikes within each phase value to allow for accurate estimation of position information across all theta phases. To ensure theta modulation, we also require the most active phase to contain at least twice the number of spikes as the least active phase.

Bayesian population decoding in Fig. 7I–L involves the same binning as in the single-neuron case above, and we enforce a minimum spike count of 30 across all phases instead of a minimum for each phase value. We do not ensure theta modulation on a single-neuron basis. We consider all sessions in which at least 5 neurons are simultaneously recorded; there are 8 valid CA3 sessions and 25 valid CA1 sessions. For each session, we compute the total activity across neurons and turn directions as a function of theta phase to determine the sparsest and densest half of phases (similarly to Eq. (14)). We then compute activities $f_i(r, \psi)$ over each half, indexed by $\psi \in \{$sparse, dense$\}$, for neurons $i$ and turn directions $r$. For each neuron, we rectify all activity values below 0.02 times its maximum.

We decode turn direction during runs along the center arm using sliding windows of width $\Delta t = 0.5$ s and stride 0.25 s. In each window at time $t$, we tabulate the population spike count $\mathbf{c}(t, \psi)$ over sparse and dense phases $\psi$. The likelihood that it arose from turn direction $r$ is

$$p(\mathbf{c}(t,\psi)|r) = \prod_i p(c_i(t,\psi)|r) \propto \left( \prod_i f_i(r,\psi)^{c_i(t,\psi)} \right) \exp\left( -\Delta t \sum_i f_i(r,\psi) \right). \quad (16)$$

This formula assumes that spikes are independent across neurons and time and obey Poisson statistics[110]. We only decode with at least 2 spikes. By Bayes's formula and assuming a uniform prior, the likelihood is proportional to the posterior probability $p(r|\mathbf{c}(t, \psi))$ of turn direction $r$ decoded from spikes $\mathbf{c}(t, \psi)$. Consider one decoding that yields $p(R)$ as the probability of a right turn. Its confidence is $|2p(R)-1|$. Its accuracy is 1 if $p(R) > 0.5$ and the true turn direction is right or if $p(R) < 0.5$ and the true direction is left; otherwise, its accuracy is 0.

### Machine learning with multilayer perceptrons

**Dataset.** We use the MNIST dataset of handwritten digits[52]. Each image $\mathbf{i}_\alpha$ is normalized by subtracting the mean value and dividing by the standard deviation across all images and pixels. In addition to its digit class label, we randomly assign a set number. We train networks on a subset of images from the train dataset. To test concept learning through digit classification, we use all held-out images from the test dataset. To test example learning through set identification, we use all

train images corrupted by randomly setting 20% of normalized pixel values to 0.

**Single-task learning.** We train a fully-connected two-layer perceptron with a hyperbolic tangent (tanh) activation function applied to each hidden layer and a softmax activation function applied to the output layer. Each hidden layer contains 50 neurons, and the output layer contains 10 neurons for digit classification and as many neurons as sets for set identification.

Let $\mathbf{s}_\alpha$ be the activations of the final hidden layer for image $\alpha$. The loss is composed of a cross-entropy loss function between reconstructed labels $\hat{\mathbf{y}}_\alpha$ and true labels $\mathbf{y}_\alpha$, which are one-hot encodings of either digit class or set number, and the DeCorr loss function:

$$\mathcal{L} = -\sum_{\substack{\alpha \in \\ \text{batch}}} \sum_{i=0}^{N-1} \left[ y_{\alpha i} \log \hat{y}_{\alpha i} + (1 - y_{\alpha i}) \log(1 - \hat{y}_{\alpha i}) \right] + \lambda \mathcal{L}_{\text{DeCorr}}, \quad (17)$$

where

$$\mathcal{L}_{\text{DeCorr}} = \frac{1}{2} \sum_{\substack{\alpha \neq \beta \in \\ \text{batch}}} \frac{\left[ \sum_{i=0}^{N-1} (s_{\alpha i} - \bar{s}_\alpha)(s_{\beta i} - \bar{s}_\beta) \right]^2}{\left[ \sum_{i=0}^{N-1} (s_{\alpha i} - \bar{s}_\alpha)^2 + N\epsilon \right] \left[ \sum_{i=0}^{N-1} (s_{\beta i} - \bar{s}_\beta)^2 + N\epsilon \right]}. \quad (18)$$

We introduce $\epsilon = 0.001$, which is scaled by the number of hidden layer neurons $N$, to aid numerical convergence. Mean activations are $\bar{s}_\alpha = (1/N) \sum_{i=0}^{N-1} s_{\alpha i}$. The DeCorr strength is $\lambda$; except for Supplementary Fig. 8A, B, we use $\lambda = 0$ for the baseline case and $\lambda = 1$ for the DeCorr case.

We train the network using stochastic gradient descent with batch size 50, learning rate $10^{-4}$, and momentum 0.9. In general, we train until the network reaches > 99.9% accuracy with the train dataset. For example, we use 40, 100, and 200 epochs respectively for digit classification and set identification with 10 and 50 sets.

In contrast to DeCorr, the DeCov loss function formulated to reduce overfitting is

$$\mathcal{L}_{\text{DeCov}} = \frac{1}{2} \sum_{i \neq j = 0}^{N-1} \left[ \sum_{\substack{\alpha \in \\ \text{batch}}} (s_{\alpha i} - \bar{s}_i)(s_{\alpha j} - \bar{s}_j) \right]^2, \quad (19)$$

where mean activations are now taken over batch items: $\bar{s}_i = (1/N_{\text{batch}}) \sum_{\alpha \in \text{batch}} s_{\alpha i}$[53].

**Multitask learning.** We train a fully-connected two-layer perceptron with a hyperbolic tangent (tanh) activation function applied to each hidden layer. In Supplementary Fig. 8E, F, we also consider applying a ReLU activation function to each hidden layer, or a ReLU to the first hidden layer and no nonlinearity to the second. The final hidden layer is fully connected to two output layers, one for digit classification and the other for set identification. A softmax activation function is applied to each layer. Each hidden layer contains 100 neurons, the concept output layer contains 10 neurons, and the example output layer contains as many neurons as sets.

The loss is composed of a cross-entropy loss function between reconstructed $\hat{\mathbf{y}}_\alpha$ and true $\mathbf{y}_\alpha$ digit labels, a cross-entropy loss function between reconstructed $\hat{\mathbf{z}}_\alpha$ and true $\mathbf{z}_\alpha$ set labels, and either the DeCorr

or HalfCorr loss function:

$$\mathcal{L} = -\sum_{\substack{\alpha \in \\ \text{batch}}} \sum_{i=0}^{N-1} \left[ y_{\alpha i} \log \hat{y}_{\alpha i} + (1 - y_{\alpha i}) \log(1 - \hat{y}_{\alpha i}) \right]$$
$$- \sum_{\substack{\alpha \in \\ \text{batch}}} \sum_{i=0}^{N-1} \left[ z_{\alpha i} \log \hat{z}_{\alpha i} + (1 - z_{\alpha i}) \log(1 - \hat{z}_{\alpha i}) \right] + \lambda \mathcal{L}_{\text{DeCorr/HalfCorr}}, \quad (20)$$

where

$$\mathcal{L}_{\text{HalfCorr}} = \frac{1}{2} \sum_{\substack{\alpha \neq \beta \in \\ \text{batch}}} \frac{\left[ \sum_{i=N/2}^{N-1} (s_{\alpha i} - \bar{s}_\alpha)(s_{\beta i} - \bar{s}_\beta) \right]^2}{\left[ \sum_{i=N/2}^{N-1} (s_{\alpha i} - \bar{s}_\alpha)^2 + N\epsilon/2 \right] \left[ \sum_{i=N/2}^{N-1} (s_{\beta i} - \bar{s}_\beta)^2 + N\epsilon/2 \right]}. \quad (21)$$

Mean activations are $\bar{s}_\alpha = (2/N) \sum_{i=N/2}^{N-1} s_{\alpha i}$. The DeCorr/HalfCorr strength is $\lambda$; we use $\lambda = 1$ with a tanh activation function, $\lambda = 0.04$ with a ReLU, $\lambda = 2$ with no nonlinearity, and $\lambda = 0$ for the baseline case with any nonlinearity.

We train the network using stochastic gradient descent with batch size 50 and learning rate $10^{-4}$. In general, we train until the network reaches > 99.9% accuracy in both tasks with the train dataset. For example, we use 100 epochs for the results in Fig. 8I, J.

### Reporting summary

Further information on research design is available in the Nature Portfolio Reporting Summary linked to this article.

## Data availability

All experimental data used in this study are taken from the Collaborative Research in Computational Neuroscience (CRCNS) hc-3 dataset contributed by György Buzsáki and colleagues[24] and the hc-6 dataset contributed by Loren Frank and colleagues[25]. They are publicly available at https://crcns.org/data-sets/hc. The MNIST dataset[52] used in this study is available at http://yann.lecun.com/exdb/mnist. The FashionMNIST dataset[26] used in this study can be found at https://github.com/zalandoresearch/fashion-mnist. Source data are provided with this paper.

## Code availability

All network training and simulation code is available at https://github.com/louiskang-group/kang-2024-examples-concepts.

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

## Acknowledgements

We thank Tom McHugh and Łukasz Kuśmierz for helpful ideas. LK is supported by JSPS KAKENHI for Early-Career Scientists (22K15209) and has been supported by the Miller Institute for Basic Research in Science and a Burroughs Wellcome Fund Collaborative Research Travel Grant. TT is supported by Brain/MINDS from AMED (JP19dm0207001) and JSPS KAKENHI (JP18H05432).

## Author contributions

L.K. and T.T. conceptualized the study, analyzed the results, and wrote the manuscript. L.K. trained and simulated the neural networks, performed the mathematical derivations, and analyzed the experimental data.

## Competing interests

The authors declare no competing interests.
