## [Peer Review File · Nature Communications]

Distinguishing examples while building concepts in hippocampal and artificial networksREVIEWER COMMENTS

Reviewer #1 (Remarks to the Author):

Kang and Toyozumi — NCOMMS-23-10468-T

Overview:

This modeling and data analysis study by Kang and Toyozumi presents a novel theoretical interpretation of the two branching inputs to hippocampal subregion CA3: indirect—via dentate granule cells and mossy fibers (MF)—and direct—via layer II entorhinal cortical neurons originating the perforant path (PP). The computational and mnemonic role(s) of these pathways have been at the center of key questions driving experimental and computational modeling research to understand hippocampal information processing, the formation of long-term episodic memory, and neural representations of place cells and cognitive maps which support spatial navigation in mammals. This manuscript connects deeply into those literatures, providing scientific context that comprehensively motivates its theoretical and computational goals of the study.

Prior models supported distinct memory encoding/retrieval roles for the dual pathways based on a single (possibly conjunctive) CA3 representation, but Kang and Toyozumi studied whether the sparse/strong MF→CA3 input and the dense/weak PP→CA3 input form distinct and separable memory encodings for, respectively, examples and concepts that may be selectively expressed by high or low network activity thresholds (such as reflected by inhibitory tone of CA3 interneuronal subnetworks, and which is modulated according to the ongoing hippocampal theta rhythm). Moreover, the authors' approach combined artificial neural network (ANN) modeling with spin-glass dynamics, in contrast to the continuous-time rate-based or spiking point-neuron circuit models that are more conventional in computational neuroscience research.

This approach puts the focus on computational function at the cost of abstracting away most biological detail. The resulting gap between the neuroscientific motivation for the study and the level of abstraction of the approach is the source of my main concerns. Overall, it's a well-written paper about an interesting but somewhat disjointed set of studies that advances computational considerations of hippocampal memory formation and translates those ideas to ANN-based machine learning models. These findings could potentially help introduce key neural mechanisms like attractor dynamics and oscillatory computing to AI/ML research.

Major concerns:

1. A paper like this should excite both neuroscientists and AI/ML researchers, because the intersection (while growing) is not yet very large. In particular, the authors' machine learning tasks and secondary analysis of hippocampal datasets are both interesting, but they serve to highlight this gap instead of building a bridge across it. Given its length and disjointedness, I encourage the authors to consider whether this manuscript should be two papers, each of which might better target its respective audience. E.g., (1) tell AI/ML engineers about the algorithmic advances and potential for performance improvements, scalability, generalization, etc.; and (2) tell (computational) neuroscientists about the new theory of hippocampal memory encoding, the threshold-based retrieval function of theta oscillations, and how your secondary data analyses support those theoretical advances. If the authors (understandably) disagree, then I would encourage a revision that better acknowledges this gap and reframes its idea and findings in an interdisciplinary way that enhances relevance and impact across both domains. This could be largely organizational (vs. major rewriting), but it should regardless be carefully considered.

2. The abstract ANN-based models of the MF and PP inputs to CA3 provide analytical tractability and applicability to AI/ML tasks, which are appropriately emphasized in the work and the manuscript. However, the design, parameterization, batch training, and analyses of the ANNs are based on many "magic numbers" sprinkled throughout the manuscript, e.g., in results, captions, methods, and supplementary information. One problem is legibility: without even a table of parameters and values, it is very difficult as a reader to put together a mental model of the systems of ANNs constructed in these studies, which includes subtleties such as how the autoencoder is used in different conditions. A second problem is robustness: the manuscript does not describe how all these values were determined. Did the authors conduct any hyperparameter optimization procedures? How many degrees of freedom do these models have in relation to the dimensionality of the problem domain? How sensitive are the findings (model results or data analyses) to changes in particular parameters? For example, the 0.1 weighting for PP vs. MF inputs in Eq. (2) seems to be important, but is it important only that $PP < MF$, or must it be close to 0.1? Ascribing parameter values of these abstract binary feedforward or "Hopfield-like" networks to "biological trends" or other vague hand-waving is poor justification. The authors should include more detail to justify their model design and training, e.g., hyperparameter optimization, regularization, cross-validation, sensitivity analyses, parametric dependences, etc. If in fact many of these values were chosen ad hoc or for convenience, then the strength of the paper's claims is undercut and these issues should be clarified in the discussion and elsewhere as appropriate.

Other concerns and comments:

* Fig. 1E: Autoencoder layer sizes typically decrease toward the middle hidden layer. Why does this network appear to have the inverse design? Is it due to the connective sparsity? If so, should this still be called an autoencoder?

* Lines 64–68: These two sentences (starting “For instance...”) are odd. First, we should be wary of labeling old ideas “classic”, which is akin to an appeal to authority on the basis of age. Second, citing Quiroga’s 2005 and 2009 papers as “recent” is quite a stretch, especially given the rate of change in neuroscience over the last 15 years. Lastly, the notion of “grandmother cells” argues against sparse distributed representations, so bringing it up as an example of generalization here appears to be a non sequitur: i.e., you could have sparse statistical or semantic generalization without denying distributed representations. Of course, if the authors in fact intended to deny that hippocampal attractor-based representations like those discussed here are not distributed, then that would be surprising and should be made explicit.

* Lines 69–81: This paragraph summarizes results from the study, so most of these verbs should be in past tense, not in present or future tenses. E.g., “We will test these predictions...” should be “We tested these predictions...”, “each analysis will reveal that...” should be “each analysis revealed that...”. This linguistic distinction clarifies for readers what you have specifically done in conducting the research supporting this paper.

* Ibid.: The last paragraph of the introduction states the main prediction in the paper as the relationship between network sparsity and sharper tuning. If we consider a Gaussian place field and a threshold, then we can imagine that this threshold moves up or down. This mental exercise makes it trivially clear that sparser/denser activation is geometrically coupled to sharper/broader spatial tuning. The authors should clarify throughout the paper how the main prediction or outcomes of their study is not simply an a priori consequence of this trivial geometric relationship.

* Line 87: “The sensory inputs that constitute memories in our model...” This must be clarified. Sensory inputs do not constitute memories in models of memory; these are very different things.

* Line 92: “...random, binary, and sparse...” Is this supposed to be respective to the DG, MF, and PP encodings? This is a good example of a sentence that raises more questions than information it provides. E.g., aren’t all of these projections binary? If all projections are random, binary, and sparse, then how and why do the parameters or distributions of randomness and sparsity vary across the projections?

* Line 98: (Addressing the authors here.) I understand that you want your variable ‘sparsity’ to map naturally to [0,1], but to define ‘sparsity’ as the inverse of the actual meaning of the English word “sparsity” is very confusing. E.g., on Line 106, you equate “decreasing sparsity” with “sparsification”, which would be defined in English as “increasing sparsity”! Then, on Line 110, you introduce ‘sparsity’ as a_{pre} , which seems to reflect an “activity” level (as in, ‘a’ for “activity”). I would strongly encourage the authors to consider refactoring this variable name from ‘sparsity’ to ‘active fraction’ or something similar.

Then, “decreasing active fraction” clearly means “less activity” and thus “sparsification”, with no need for mind-bending mental translation on the part of your readers.

* Line 118–124: To solve the problem of translating CA3 patterns back to images, the authors train a continuous-valued feedforward EC→CA3 network to feed CA3 output to the decoder half of the autoencoder. This seems like a very complicated way to include a CA1 network. The unexplained complexity here raises a number of questions. Why is this image-generating network continuous-valued, but the main EC→CA3 networks are binary? Why not add a CA1 module and train the whole trisynaptic loop? Is this effectively the same? Is this network only for research visualization purposes? Wouldn't the brain need to convert CA3 output into reconstructed memories anyway? So why not study this additional network in the same context? Does the decodability of the two encodings rely on an assumption of disentangled representations, e.g., via linear superposition?

* Line 127: The assumption of linear dendritic integration between proximal and distal sites is simply false (cf. Mel, Poirazi, et alia). If the relative weighting of 0.1 PP vs. 0.9 MF is based on this assumption, then the resultant findings of threshold-based effects that depend on that weighting and its linearity may not hold up to scrutiny. The authors should provide additional context for understanding this assumption and its follow-on implications in the results and discussion sections.

* Line 157: It's unclear whether the authors (or others in the field) might have expected a “concept” representation to reflect a simple average of a set of exemplars. This seems like a surprising expectation to me and potentially underlies the primary distinction this paper is making between “examples” and “concepts”. It is not difficult to think of many reasons why an average example may not serve the required abstractions and other functions of a concept. This relationship between examples and concepts should be clarified.

* Fig. 3C: Pie charts are difficult to parse for many people. Please consider using bar or stacked charts instead.

* Fig. 4D: This shows the trivial geometric relationship between sparsity and tuning that I described above. The “alternative prediction” in the right-hand panel does not convey equivalent empirical content as the “model prediction” on the left, and is thus much easier to dismiss. Treating these two possibilities as equally likely is equivalent to overclaiming the statistical power of your inference. If this is not the case, then the authors should clarify why the left panel does not reflect a trivial relationship and the right reflects an equally likely possibility given what we (in the field) already know about how place cells fire as a rodent traverses a place field.

* Line 273: Use of 4 spatial bins is confounded with place-field size, so the relative offset of the bins should have also been considered to avoid field-edge effects. The authors should clarify and justify how they determined bin numbers, sizes, and offsets to compute spatial information of place-cell firing.

* Line 391: “experimental findings” → “secondary analysis of experimental data”

* Line 410: In discussing Skaggs (1996), the authors compare hypothetical results after mentioning differences in binning technique and offsets. Instead of hand-waving and speculating, why not go ahead and replicate the Skaggs binning technique so you can show us whether or how much it matters? Additionally, the authors neglected to discuss another highly relevant and more recent study that likewise conducted a secondary analysis of the hc-3 dataset: Souza & Tort. Asymmetry of the temporal code for space by hippocampal place cells. *Scientific Reports*. 2017;7(1):8507.

* Line 480: N=256 images seems like a very small trainset. The authors should provide additional context and justification for low-sample training, including their restriction to a small number of categories in the FashionMNIST dataset. Substantial downselection of the trainset could impact the generalizability and robustness of model inference based on these ANNs.

* Line 524: The authors state that cues are formed by flipping only 1% of all neurons in a given target pattern. From this, should I infer that the primary pattern completion inference problem in this the study is to correct a cue that is already a 99% match to a pre-trained target pattern? Is the network being initialized to a state that is 99% of the way to a correct answer? If this is the case, then I fail to see how the problem-solving performance of these network models can be assessed given the data presented.

Reviewer #2 (Remarks to the Author):

The authors proposed a computational model for EC, DG and CA3 in which CA3, a Hopfield-like network, stores both correlated and decorrelated encoding patterns. These two patterns can be retrieved respectively at low and high inhibitory tone. When the inhibitory threshold is high, sparser MF patterns are more likely to be recovered, while when it is low, denser PP patterns are more likely to be recovered. To experimentally test for the presence of these complementary encodings, the authors further analyzed theta modulated tuning of hippocampal CA3 neurons, and revealed that these neurons alternate between finer example-like representations and broader, concept-like representations of space across individual theta cycles. Finally, the authors generalized the hippocampal circuit model to an artificial neural network and demonstrated that the CA3-like complementary encodings improve neural network performance in multitask machine learning. Overall, I found this paper to be a fascinating and insightful

exploration of the complementary encoding strategies employed by the hippocampal circuit. The proposed computational model is both elegant and intuitive, and their experimental findings provide compelling evidence for the existence of these distinct encoding modes. It is interesting that the authors demonstrated that these computational properties found in hippocampal circuits could be applied to improve machine learning performance. This paper represents an important contribution to our understanding of the neural mechanisms underlying learning and memory.

Below are the major/minor points I would love the authors to address.

Major points:

1, The retrieval of PP patterns appears similar to the average image in one category, and the authors treated this as retrieving a concept of that category. Can the authors provide more evidence about why this mean image can represent the concept of a 'concept'? Is this mean image an analysis artifact due to the simple dataset where images belonging to a single category do not have large variance, so the averaged image can still be recognized as sneakers, trousers, or coats. In other words, the authors could provide more evidence for why the mean image can represent the concept of a category by discussing previous studies that have used similar methods and how they have validated their approach. They could also address the concern about the simple dataset used and discuss whether the mean image approach would still hold for more complex datasets (such as Cifar10).

2, The threshold θ in the sigmoid function is controlled by the width of the function, but how is the changing of the width being linked to the hippocampal theta oscillation? More importantly, the oscillating threshold does not necessarily represent the oscillation rhythm at the theta band. The authors might want to look at the Pfeiffer and Foster (2015) paper where they found the slow gamma oscillations route the information flow during offline states, i.e., the decoded trajectory from awake SWRs.

Minor points:

1, Pattern separation through the MF pathway might take time. In this case, will the MF and PP encodings arrive at CA3 at the same time? If one arrives earlier than the other, will it still be a simple addition in the mathematical model?

2, The authors could explain why they chose to use the autoencoder to represent the neural representation of EC neurons and provide some evidence for why this approach is valid.

3, To my understanding, the synaptic connections in the Hopfield-like network (Eq.2) will give attractor states as the superimposed patterns, e.g., $0.9x_i^{\text{MF}} + 0.1x_i^{\text{PP}}$. How can the network retrieve either x_i^{MF} or x_i^{PP} ?

4, It will be clearer if simply do a histogram plot of the spike numbers against LFP theta phase to show the theta modulation of place cell activity (line 211 and Fig.4B, making the histogram on the right side of panel B bigger).

5, Is the Wilcoxon signed-rank test in Fig.4K two-tailed or one-tailed?

Overall, the authors provided compelling evidence for the existence of complementary encoding modes in CA3, and their findings shed new light on the neural mechanisms underlying learning and memory. If the authors could address the issues I raised, particularly the major points related to the concept representation and the link between the threshold and hippocampal theta oscillation, then I would be happy to recommend this paper for publication.

Reviewer #3 (Remarks to the Author):

The authors present a paper arguing first that different pathways converging upon hippocampal area CA3 may preferentially code for more episode-unique (DG-CA3) and more conceptual (EC-CA3) information. Next, they show evidence for an alternation of biases across the hippocampal theta rhythm towards sparser (more episodic) and then coarser (more conceptual) information in CA3 place cells, which was highly impressive. Finally, they show this type of parallel coding can improve learning in a more general neural network architecture. I found the paper impressive overall in its ambition and scope. It was well-written and well-argued, with very few mistakes or unclear passages, and it was very thorough in terms of having strong primary findings as well as nice ancillary findings. Here are some comments that I hope the authors will consider:

Intro/Discussion: Theta power changes during various tasks [e.g., increases at successful encoding (Jacobs et al., 2006; numerous Michael Kahana papers) or exploration in rodents (countless papers)]. Does theta power mean more sparsity overall or less? Or just a wider range of sparsity levels across the cycle? Addressing this point could link these findings to vast literatures that largely report theta power values, and it could generate new hypotheses.

Figure 1 – Generally, D-H could use better organization / illustration. For instance, where does the right half of E fit in with respect to the architecture in D? Can you visually label where in D that F fits in (along EC-CA3)? Or is F more conceptual than the specific hippocampal model, of which EC-CA3 is only one case (as I believe they are indicating in G)?

Smaller points

Abstract – first sentence:

Using ‘sensory’ makes it seem as if the hippocampus does not store ‘conceptual’ information along with a memory trace.

“Meanwhile, recent research has found that the hippocampus also participates in semantic memory, as evidenced by grandmother cells that generalize over your visits and respond to many different representations of your grandmother (Quiroga et al., 2005, 2009).” Norman et al. (2021, Science) is also relevant to this point.

p. 4 – “Sparsity is the fraction of active neurons, so lower values correspond to sparser patterns.” As a concept, usually higher values of that concept indicate more of that concept, so, generally speaking, higher “sparsity” values would mean more sparse patterns rather than less sparse ones. Ultimately, I appreciate that the authors define this, and I respect their choice if they decide to keep this term, but perhaps they could consider using the opposite term, “density”.

End of p. 4 – It may help to orient the reader to point out that after CA1, the signals would need to get back out to cortex.

Fig 2G – Does the PP concepts line go higher than the maximum achievable overlap? If so, is this an error, or have I misunderstood?

Top p. 6 – “because since” typo

p. 6 – “During retrieval, the network is asynchronously updated via Glauber dynamics (Amit et al., 1985). That is, at each simulation timestep, one neuron is randomly selected to be updated

(Fig. 2C). If its total input from other neurons exceeds a threshold θ , then it is more likely to become active.” In this scheme, is there only further activation, or do active neurons also become randomly selected to possibly become inactivated (balancing the overall activation level)?

Bottom p. 6 to p. 7 – For myself as a reader, these paragraphs were incredibly helpful – I think the authors really hammer the main points home here well.

p. 10 – I very much appreciate this alternative prediction from the attention literature. It clarifies the need to resolve it via data, as they do below.

Discussion:

By my reading, Kowadlo et al. (2019, arXiv) & Antony et al. (2022, bioRxiv) present computational models that also argue for some generalization occurring in the PP (EC-CA3) pathway. The authors should consider how their model's predictions support or differ from these papers.

(Addressing this point is optional.) My broadest question is how would these different theta phases playing different roles in both encoding and retrieval play out in the subjective encoding or retrieval of a memory? Do the authors believe a rodent or human can meaningfully alter (or pay attention to) the sparsity of their representations at the rate of a theta rhythm? If one were asked to recall the specific or conceptual associations to a given cue, would a control mechanism (say, from the prefrontal cortex) properly tune the theta rhythm in some way to accomplish this task? I think speculation along these lines would be helpful if the authors have insight, but if it is too speculative, it's fine to omit.

Methods:

Line 527 – “expect” -> “except”

12 June 2023

Dear *Nature Communications* Editors and Reviewers:

Below we explain how our new manuscript has addressed all the Reviewers' comments and is consequently stronger, clearer, and better connected with previous literature. Changes in the main text have been indicated in red, and all figure and line numberings invoked in this response refer to the revised version of the manuscript.

Reviewer 1

Major point 1: “A paper like this should excite both neuroscientists and AI/ML researchers, because the intersection (while growing) is not yet very large. In particular, the authors' machine learning tasks and secondary analysis of hippocampal datasets are both interesting, but they serve to highlight this gap instead of building a bridge across it. Given its length and disjointedness, I encourage the authors to consider whether this manuscript should be two papers, each of which might better target its respective audience. E.g., (1) tell AI/ML engineers about the algorithmic advances and potential for performance improvements, scalability, generalization, etc.; and (2) tell (computational) neuroscientists about the new theory of hippocampal memory encoding, the threshold-based retrieval function of theta oscillations, and how your secondary data analyses support those theoretical advances. If the authors (understandably) disagree, then I would encourage a revision that better acknowledges this gap and reframes its idea and findings in an interdisciplinary way that enhances relevance and impact across both domains. This could be largely organizational (vs. major rewriting), but it should regardless be carefully considered.”

We agree with the Reviewer on the importance of a strong connection between our neuroscience and ML components, and we welcome this opportunity to tie the two together more tightly. First, we reorganized the first figure to focus on the overarching organization of the entire work, with a new Fig. 1C that schematically illustrates how its components tie together. The original manuscript conveyed how our investigation into CA3 encodings inspired the HalfCorr loss function, but it did not sufficiently explain how HalfCorr networks allow us to probe the function of CA3-like encodings on explicit learning tasks. Thus, we introduced new text in lines 77 and 510 emphasizing the point that training artificial networks enables us to study whether CA3-like encodings are indeed good for complementary computational tasks. With these changes, we feel that our manuscript not only motivates how our neuroscience results contribute to ML developments, but also how our ML results extend the significance of our neuroscience findings.

Major point 2: “The abstract ANN-based models of the MF and PP inputs to CA3 provide analytical tractability and applicability to AI/ML tasks, which are appropriately emphasized in the work and the manuscript. However, the design, parameterization, batch training, and analyses of the ANNs are based on many “magic numbers” sprinkled throughout the manuscript, e.g., in results, captions, methods, and supplementary information. One problem is legibility: without even a table of parameters and values, it is very difficult as a reader to put together a mental model of the systems of ANNs constructed in these studies, which includes subtleties such as how the autoencoder is used in different conditions. A second problem is robustness: the manuscript does not describe how all these values were determined. Did the authors conduct any hyperparameter optimization procedures? How many degrees of freedom do these models have in relation to the dimensionality of the problem domain? How sensitive are the findings (model results or data analyses) to changes in particular parameters? For example, the 0.1 weighting for PP vs. MF inputs in Eq. (2) seems to be important, but is it important only that $PP < MF$, or must it be close to 0.1? Ascribing parameter values of these abstract binary feedforward or “Hopfield-like” networks to “biological trends” or other vague hand-waving is poor justification. The

authors should include more detail to justify their model design and training, e.g., hyperparameter optimization, regularization, cross-validation, sensitivity analyses, parametric dependences, etc. If in fact many of these values were chosen ad hoc or for convenience, then the strength of the paper’s claims is undercut and these issues should be clarified in the discussion and elsewhere as appropriate.”

This is a good point. Our parameter choices were guided by theoretical results for networks that store random MF and PP patterns (Kang & Toyozumi, arXiv:2302.04481, 2023), and we did not need to perform extensive fine-tuning. To confirm the generality of our results with respect to parameter values, we now present an extensive parametric exploration (Fig. S3F), encompassing PP pattern storage strength, pattern densities, sources of cue noise, and dendritic nonlinearity. Over wide ranges of each parameter, our CA3 model maintains its central capability of recovering both MF examples and PP concepts from both MF and PP cues. This supports the robustness of our network and argues against dependence on “magic numbers.” Also, we now summarize our key model parameters in Table 1 in the Methods section.

Minor point 1: “Fig. 1E: Autoencoder layer sizes typically decrease toward the middle hidden layer. Why does this network appear to have the inverse design? Is it due to the connectional sparsity? If so, should this still be called an autoencoder?”

We wanted our EC network to be large to increase capacity and reduce finite-size effects, especially because our EC patterns are sparse. While it’s true that many autoencoders implement a compressed middle layer for dense feature extraction, overcomplete autoencoders with an expanded middle layer are also used for unsupervised feature extraction, especially in the context of sparse coding. In fact, sparse, overcomplete networks are commonly used to model the encoding of natural scenes (Olshausen & Field, 1996; Makhzani & Frey, 2014; etc.). We have clarified these points in line 95.

Minor point 2: “Lines 64–68: These two sentences (starting “For instance...”) are odd. First, we should be wary of labeling old ideas “classic”, which is akin to an appeal to authority on the basis of age. Second, citing Quiroga’s 2005 and 2009 papers as “recent” is quite a stretch, especially given the rate of change in neuroscience over the last 15 years. Lastly, the notion of “grandmother cells” argues against sparse distributed representations, so bringing it up as an example of generalization here appears to be a non sequitur: i.e., you could have sparse statistical or semantic generalization without denying distributed representations. Of course, if the authors in fact intended to deny that hippocampal attractor-based representations like those discussed here are not distributed, then that would be surprising and should be made explicit.”

We have added explicit references for episodic memory instead of using the term “classic,” and we have removed the word “recent.” We do not intend to argue against distributed representations in this passage, and we do not feel that the work of Quiroga and colleagues provides strong evidence against the possibility of distributed representations. For example, in Quiroga et al. (2005), only 43 out of 132 responsive units demonstrated specific responses to a single entity. This may be higher than chance level, but both neurons with singular tuning and those with more numerous responses are present. Moreover, “grandmother cells” may also code memories in a distributed fashion, with the possibilities of multiple units tuned to the same entity and singularly tuned units responding to other stimuli not tested—132 entities is still small compared to the enormous range of individuals and objects remembered over a lifetime. While we model memories as fully distributed in our Hopfield-like network, the biological system probably lies somewhere between a one-to-one coding of memories and a completely distributed coding with each neuron’s participation determined randomly and independently.

Minor point 3: “Lines 69–81: This paragraph summarizes results from the study, so most of these verbs should be in past tense, not in present or future tenses. E.g., “We will test these predictions...” should be “We tested these predictions...”,”

“each analysis will reveal that...” should be “each analysis revealed that...”. This linguistic distinction clarifies for readers what you have specifically done in conducting the research supporting this paper.”

We appreciate this suggestion and have made changes accordingly. We use the past tense to describe explicit actions, such as constructing models and testing predictions. We use the present tense to describe the consequences of these actions, which are not tied to one moment in time. We hope that these guidelines are sensible to the Reviewer.

Minor point 4: “Ibid.: The last paragraph of the introduction states the main prediction in the paper as the relationship between network sparsity and sharper tuning. If we consider a Gaussian place field and a threshold, then we can imagine that this threshold moves up or down. This mental exercise makes it trivially clear that sparser/denser activation is geometrically coupled to sharper/broader spatial tuning. The authors should clarify throughout the paper how the main prediction or outcomes of their study is not simply an a priori consequence of this trivial geometric relationship.”

Indeed, the proposed function of the theta oscillation as an activity threshold is central to our model. While its effect on tuning appears obvious, to our knowledge, we do not know of any work that explicitly demonstrates that theta operates in this fashion. In lines 245 and 417, we set our work in a broader context by discussing subtractive and divisive inhibition, both of which have been observed throughout cortical networks. Our theta model argues for a subtractive role on firing rates, but we believe that a priori, the alternative hypothesis of a divisive role is equally valid. In fact, hippocampal pyramidal cells have been observed to experience divisive inhibition on their membrane potentials (Losonczy et al., 2010; Bhatia et al., 2019); downstream effects on firing rates have not been explored. Thus, although some readers may intuitively expect our model prediction, investigation into its consequences for CA3 tuning, especially with respect to the retrieval of different memory encodings, has not been as thoroughly performed before to our knowledge.

Minor point 5: “Line 87: “The sensory inputs that constitute memories in our model...” This must be clarified. Sensory inputs do not constitute memories in models of memory; these are very different things.”

We have clarified that the encodings are stored as memories, not the sensory inputs themselves, by changing this phrase to “The sensory inputs whose encodings serve as memories in our model are FashionMNIST images....”

Minor point 6: “Line 92: “...random, binary, and sparse...” Is this supposed to be respective to the DG, MF, and PP encodings? This is a good example of a sentence that raises more questions than information it provides. E.g., aren’t all of these projections binary? If all projections are random, binary, and sparse, then how and why do the parameters or distributions of randomness and sparsity vary across the projections?”

We have now made explicit the connections that are modeled by these types of matrices (line 101). Yes, the EC  DG, DG  MF, and EC  PP projections are all binary. We have added a phrase at the end of the Introduction section to alert the reader that parameter descriptions, justifications, and values are found in the Methods section and Table 1 (line 82). Across projections, synapses are always randomly chosen such that each postsynaptic neuron receives a fixed number of excitatory synapses from presynaptic neurons. These fixed numbers, or projection sparsities, are chosen to follow biological trends; due to uncertainties in experimental measurements and in the proper way to rescale biological counts for model networks with much fewer neurons, precise determination of these sparsities is not possible. We have also explicitly mentioned in the Methods section that based on our theory, we do not expect projection sparsity to be a crucial parameter (line 566). We hope that these changes improve the understanding of our model.

Minor point 7: “Line 98: (Addressing the authors here.) I understand that you want your variable ‘sparsity’ to map naturally to [0,1], but to define ‘sparsity’ as the inverse of the actual meaning of the English word “sparsity” is very confusing. E.g., on Line 106, you equate “decreasing sparsity” with “sparsification”, which would be defined in English as “increasing sparsity”! Then, on Line 110, you introduce ‘sparsity’ as a_{pre} , which seems to reflect an “activity” level (as in, ‘a’ for “activity”). I would strongly encourage the authors to consider refactoring this variable name from ‘sparsity’ to ‘active fraction’ or something similar. Then, “decreasing active fraction” clearly means “less activity” and thus “sparsification”, with no need for mind-bending mental translation on the part of your readers.”

We appreciate the Reviewer’s suggestion, which is echoed in Reviewer 3’s minor point 3, and we use Reviewer 3’s term “density” to describe the variable a .

Minor point 8: “Line 118–124: To solve the problem of translating CA3 patterns back to images, the authors train a continuous-valued feedforward EC-CA3 network to feed CA3 output to the decoder half of the autoencoder. This seems like a very complicated way to include a CA1 network. The unexplained complexity here raises a number of questions. Why is this image-generating network continuous-valued, but the main EC-CA3 networks are binary? Why not add a CA1 module and train the whole trisynaptic loop? Is this effectively the same? Is this network only for research visualization purposes? Wouldn’t the brain need to convert CA3 output into reconstructed memories anyway? So why not study this additional network in the same context? Does the decodability of the two encodings rely on an assumption of disentangled representations, e.g., via linear superposition?”

Yes, the decoding network is for visualization only. We clarify this point in line 123 and with a new label in Fig. 2B. Indeed, the brain would need to convert CA3 outputs back into reconstructed memories. However, we find that modelling this reconstruction process is beyond the scope of our manuscript. To model outputs through CA1, one would need to also consider the temporoammonic projection from EC back to CA1, as well as additional projections from CA1 to subiculum, from subiculum to EC, and from EC to subiculum. Thus, carefully making claims about CA1 would require a much more complex model than the one we have constructed to study CA3. Since we do not intend to model CA1, we use continuous-valued neurons. We expect that using binary neurons would not significantly affect the decoding performance of the network, especially with an increased number of hidden layer neurons, as supported by many studies on quantization in neural networks (Hubara et al., Adv. NIPS, 2016; etc.).

Minor point 9: “Line 127: The assumption of linear dendritic integration between proximal and distal sites is simply false (cf. Mel, Poirazi, et alia). If the relative weighting of 0.1 PP vs. 0.9 MF is based on this assumption, then the resultant findings of threshold-based effects that depend on that weighting and its linearity may not hold up to scrutiny. The authors should provide additional context for understanding this assumption and its follow-on implications in the results and discussion sections.”

We now discuss nonlinear dendritic integration more extensively (line 134) and perform simulations that implement it (Fig. S3F). With binary dendritic inputs, only one type of nonlinearity is possible: given the strength of, say, 0.1 for an active PP input and 0.9 for an active MF input, what is the strength when both are active? It is either 1 (linear), less than 1 (sublinear), or greater than 1 (superlinear). With both sublinear and superlinear values, our network can still function (Fig. S3F), demonstrating that our assumption of linear integration is not crucial to our results.

Minor point 10: “Line 157: It’s unclear whether the authors (or others in the field) might have expected a “concept” representation to reflect a simple average of a set of exemplars. This seems like a surprising expectation to me and potentially underlies the primary distinction this paper is making between “examples” and “concepts”. It is not difficult to think of many reasons why an average example may not serve the required abstractions and other functions of a concept. This relationship between examples and concepts should be clarified.”

This concern is also shared in Reviewer 2's major point 1. We have now provided additional information at lines 170 and 178, as well as a new Fig. S2A. This figure shows that examples from each concept form well-separated clusters in image space. Thus, mean images, which lie at their centroids, can serve as concept representations. We have also discussed that with more complex datasets, the mean image approach may not work, and we would suggest averaging features of a trained classifier or a variational autoencoder (line 181).

Minor point 11: "Fig. 3C: Pie charts are difficult to parse for many people. Please consider using bar or stacked charts instead."

We have made this change.

Minor point 12: "Fig. 4D: This shows the trivial geometric relationship between sparsity and tuning that I described above. The "alternative prediction" in the right-hand panel does not convey equivalent empirical content as the "model prediction" on the left, and is thus much easier to dismiss. Treating these two possibilities as equally likely is equivalent to overclaiming the statistical power of your inference. If this is not the case, then the authors should clarify why the left panel does not reflect a trivial relationship and the right reflects an equally likely possibility given what we (in the field) already know about how place cells fire as a rodent traverses a place field."

Please see our response to minor point 4.

Minor point 13: "Line 273: Use of 4 spatial bins is confounded with place-field size, so the relative offset of the bins should have also been considered to avoid field-edge effects. The authors should clarify and justify how they determined bin numbers, sizes, and offsets to compute spatial information of place-cell firing."

For Fig. 6, we intentionally did not avoid splitting up fields with our partitions because the opposite scenario was adopted in Fig. 5, where intact place fields were extracted. Thus, we included all offsets that met minimum spiking requirements, as described in Methods. We added text to line 302 that clarifies our motivations. As for the bin number, distinguishing among 4 quadrants of the track seemed to reasonably capture the broadest position representation that has behavioral relevance. Note that bin number must be preserved across scales to avoid bias (Fig. S6F–H). At small scales, distinguishing between fewer bins, say only 2 adjacent bins of scale 1/16, seemed to be a too narrow information metric. Meanwhile, estimating information with more bins requires more spikes and would decrease the number of valid neurons. Balancing these two factors led us to choose 4 bins.

Minor point 14: "Line 391: "experimental findings" → "secondary analysis of experimental data""

We have made this substitution.

Minor point 15: "Line 410: In discussing Skaggs (1996), the authors compare hypothetical results after mentioning differences in binning technique and offsets. Instead of hand-waving and speculating, why not go ahead and replicate the Skaggs binning technique so you can show us whether or how much it matters? Additionally, the authors neglected to discuss another highly relevant and more recent study that likewise conducted a secondary analysis of the hc-3 dataset: Souza & Tort. Asymmetry of the temporal code for space by hippocampal place cells. Scientific Reports. 2017;7(1):8507."

The Skaggs et al. (1996) binning technique is designed for 2D environments, whereas the data we analyzed were all taken in 1D environments. Of course, it is possible to translate this technique to linear tracks, but

we do not feel that the modified comparison would provide sufficient benefit. We appreciate the Souza & Tort (2017) reference and now discuss it briefly in line 446.

Minor point 16: “Line 480: N=256 images seems like a very small trainset. The authors should provide additional context and justification for low-sample training, including their restriction to a small number of categories in the FashionMNIST dataset. Substantial downselection of the trainset could impact the generalizability and robustness of model inference based on these ANNs.”

We store a relatively small amount of examples (up to 100 in a network) within a relatively small number of concepts (3) because our CA3 network is relatively small with only 2048 neurons. Moreover, the crucial transition from recovering PP examples to recovering PP concepts can be observed within this explored range. With larger networks, more examples and concepts can be used. In Fig. 3H, I; Fig. S3G, H; and Fig. S4, we use larger networks that store random MF and PP patterns to demonstrate that our network behaviors can generalize. We hope that our new emphasis of this point in line 204 argues for the generalizability of our results.

Minor point 17: “Line 524: The authors state that cues are formed by flipping only 1% of all neurons in a given target pattern. From this, should I infer that the primary pattern completion inference problem in this the study is to correct a cue that is already a 99% match to a pre-trained target pattern? Is the network being initialized to a state that is 99% of the way to a correct answer? If this is the case, then I fail to see how the problem-solving performance of these network models can be assessed given the data presented.”

We originally chose to flip only 1% of all neurons because MF patterns have a density of 2%, so this process adds noise whose quantity is 50% of the original pattern, which can be considered the signal. However, it is true that 1% flipping is much less significant for PP patterns with a density of 20%. Moreover, for both MF and PP patterns, only 1% the active neurons in the target pattern are inactivated, so the signal largely remains intact. To address this good point, we perform simulations in which more neurons are flipped; recovery of both MF examples and PP concepts from both MF and PP cues can be largely maintained up to 10% of random flipping of all neurons, active and inactive (cue inaccuracy in Fig. S3F). Instead of random flipping, we also consider randomly inactivating a fraction of the active neurons to form cues (cue incompleteness in Fig. S3F). Up to 40% inactivation of the active neurons can be tolerated without a considerable decline in performance. We have also added accompanying text in lines 190 and 586.

Reviewer 2

Major point 1: “The retrieval of PP patterns appears similar to the average image in one category, and the authors treated this as retrieving a concept of that category. Can the authors provide more evidence about why this mean image can represent the concept of a ‘concept’? Is this mean image an analysis artifact due to the simple dataset where images belonging to a single category do not have large variance, so the averaged image can still be recognized as sneakers, trousers, or coats. In other words, the authors could provide more evidence for why the mean image can represent the concept of a category by discussing previous studies that have used similar methods and how they have validated their approach. They could also address the concern about the simple dataset used and discuss whether the mean image approach would still hold for more complex datasets (such as Cifar10).”

We have clarified this point now with additional information at lines 170 and 178, as well as a new Fig. S2A. This figure confirms the Reviewer’s intuition that examples from each concept form well-separated clusters in image space. Thus, mean images, which lie at their centroids, can serve as concept

representations. With more complex datasets, the mean image approach may not work, and we would suggest averaging features of a trained classifier (line 181). Alternatively, more elaborate unsupervised dimensionality reduction techniques such as variational autoencoders may extract features appropriate for concept formation. Here we used a simple encoder and decoder to study recall properties of an associative memory network.

Major point 2: “The threshold θ in the sigmoid function is controlled by the width of the function, but how is the changing of the width being linked to the hippocampal theta oscillation? More importantly, the oscillating threshold does not necessarily represent the oscillation rhythm at the theta band. The authors might want to look at the Pfeiffer and Foster (2015) paper where they found the slow gamma oscillations route the information flow during offline states, i.e., the decoded trajectory from awake SWRs.”

We regret to have misinformed the Reviewer; the threshold θ is the position of the center of the sigmoid function. A higher threshold requires each neuron to receive more recurrent excitatory input in order to fire; thus, it models a general inhibitory tone applied to all principal cells. As the Reviewer may already appreciate, the theta oscillation is believed to serve as this inhibitory tone, since medial septum inputs that drive the oscillation modulate the inhibitory interneurons in CA3. To prevent this confusion, we have removed the label “threshold softness” from Fig. 3C.

The width of the sigmoid function is the temperature of the Glauber dynamics (Eq. 11), which contributes a degree of randomness during pattern retrieval. It models noise in biological neurons, which may not always fire when their membrane potential exceeds a fixed threshold value by a small amount. This temperature is fixed for all our simulations.

The possibility that a different oscillation can serve as the varying threshold in our model during offline states is very interesting. We thank the Reviewer for this reference, and we now discuss this possibility in line 450.

Minor point 1: “Pattern separation through the MF pathway might take time. In this case, will the MF and PP encodings arrive at CA3 at the same time? If one arrives earlier than the other, will it still be a simple addition in the mathematical model?”

We consider this point in line 468. With the symmetric Hebbian STDP in our model, small timing differences would not be consequential, but with an asymmetric temporal component, PP-to-MF connections could be stronger than MF-to-PP connections, which would presumably favor the heteroassociation from PP concepts to MF examples. However, we note that if each memory’s encodings are presented tonically to CA3—over the course of a second, for example—then the short temporal difference of tens of milliseconds due to transmission through DG would not have a significant impact.

Minor point 2: “The authors could explain why they chose to use the autoencoder to represent the neural representation of EC neurons and provide some evidence for why this approach is valid.”

In line 95, we now explain that we follow previous works that modeled the neural encoding of natural scenes with sparse, overcomplete coding models and autoencoders (Olshausen & Field, 1996; Makhzani & Frey, 2014; etc.). While our images are not natural scenes per se and EC lies many layers away, so to speak, from V1 in the visual pathway, a sparse, overcomplete autoencoder provides many desired features for our model: tunable sparsity, a large number of EC neurons to reduce finite-size effects in our simulations, and the learning of memories in an unsupervised way. It is true that other architectures, such as deep convolutional neural networks for image classification (Yamins & DiCarlo, 2016; etc.) have also

been invoked to study visual pathways. However, we felt that this supervised approach would conflict with the unsupervised building of concepts that we wanted to demonstrate in our model.

Minor point 3: “To my understanding, the synaptic connections in the Hopfield-like network (Eq.2) will give attractor states as the superimposed patterns, e.g., $0.9x_i^{MF} + 0.1x_i^{PP}$. How can the network retrieve either x_i^{MF} or x_i^{PP} ?”

Indeed our network attempts to retrieve something like the linear combination $0.9 x^{MF} + 0.1 x^{PP}$; however, it consists of binary neurons, so values between 0 and 1 are not accessible. At high threshold, neural activity is disfavored, so only the most strongly stored component would be recovered: x^{MF} . As the threshold is lowered, more neurons are encouraged to activate, so neurons participating in the less strongly stored component would also activate: x^{PP} . Thus, the recovered pattern would encompass active neurons in either x^{MF} or x^{PP} , all of which take the same activity value 1. Since x^{MF} is much sparser than x^{PP} , this combined pattern is very similar to x^{PP} , so x^{PP} is approximately recovered at low threshold. We have additional explanation to line 158 which we hope makes this process clearer.

Minor point 4: “It will be clearer if simply do a histogram plot of the spike numbers against LFP theta phase to show the theta modulation of place cell activity (line 211 and Fig.4B, making the histogram on the right side of panel B bigger).”

We take this good suggestion and plot histograms for 5 cells in Fig. 5C.

Minor point 5: “Is the Wilcoxon signed-rank test in Fig.4K two-tailed or one-tailed?”

The test is two-tailed, as is now specified for all Wilcoxon signed-rank tests and Mann-Whitney U tests.

Reviewer 3

Major point 1: “Intro/Discussion: Theta power changes during various tasks [e.g., increases at successful encoding (Jacobs et al., 2006; numerous Michael Kahana papers) or exploration in rodents (countless papers)]. Does theta power mean more sparsity overall or less? Or just a wider range of sparsity levels across the cycle? Addressing this point could link these findings to vast literatures that largely report theta power values, and it could generate new hypotheses.”

We thank the Reviewer for pointing us to this line of research, with which we now engage in our manuscript. As the amplitude of an oscillation, we believe that theta power would be most likely related to the range of sparsity levels, which determines the ease with which both example and concept encodings can be accessed. We perform new simulations to touch upon this topic; to study theta power during memory retrieval, we change the amplitude of our threshold oscillation (Fig 4A, C). We find that with reduced oscillation amplitude, the network lingers on sparse example encodings instead of alternating with dense concept encodings. And by also changing the oscillation midpoint, it can linger on a dense concept (not shown). Ultimately, a more physiologically accurate model would be required to elucidate the role of theta power, whose modulation is strongly connected to changes in firing rates. We use neurons with only two states in our model. Our new simulation results are complemented by an extended discussion (line 474) that not only covers the points mentioned in this response, but also addresses possible roles of theta power during memory encoding, which was also mentioned by the Reviewer.

Major point 2: “Figure 1 – Generally, D-H could use better organization / illustration. For instance, where does the right half of E fit in with respect to the architecture in D? Can you visually label where in D that F fits in (along EC-CA3)?”

Or is F more conceptual than the specific hippocampal model, of which EC-CA3 is only one case (as I believe they are indicating in G)?”

To make the diagrams clearer, we have first separated the former Fig. 1 into Figs. 1 and 2, such that Fig. 2 can focus on the model. Furthermore, we have modified Fig. 2B such that it contains all network pathways, encoding and decoding. We then use colors and dashing to unambiguously label each pathway in both Fig. 2B and Fig. 2C–F. We have also inserted text in line 101 that hopefully improves understanding for Fig. 2D, E. Indeed, Fig. 2D, E refers to the projections from EC to DG, from DG to MF, and from EC to PP. In Fig. 2E, we also consider randomly generated x'pre instead of using x'EC to more comprehensively understand these projections on a conceptual level.

Minor point 1: “Abstract – first sentence: Using ‘sensory’ makes it seem as if the hippocampus does not store ‘conceptual’ information along with a memory trace.”

We have replaced “sensory information” with “experiences.”

Minor point 2: ““Meanwhile, recent research has found that the hippocampus also participates in semantic memory, as evidenced by grandmother cells that generalize over your visits and respond to many different representations of your grandmother (Quiroga et al., 2005, 2009).” Norman et al. (2021, Science) is also relevant to this point.”

We thank the author for this reference; we found the Norman et al., Neuron (2021) paper on semantic memory, and we now cite it in the Discussion along with other references on semantic memory (line 405).

Minor point 3: “p. 4 – “Sparsity is the fraction of active neurons, so lower values correspond to sparser patterns.” As a concept, usually higher values of that concept indicate more of that concept, so, generally speaking, higher “sparsity” values would mean more sparse patterns rather than less sparse ones. Ultimately, I appreciate that the authors define this, and I respect their choice if they decide to keep this term, but perhaps they could consider using the opposite term, “density”.”

We appreciate the Reviewer’s suggestion, which echoes Reviewer 1’s minor point 7, and use the term “density” for the variable a.

Minor point 4: “End of p. 4 – It may help to orient the reader to point out that after CA1, the signals would need to get back out to cortex.”

We now note this point in line 129.

Minor point 5: “Fig 2G – Does the PP concepts line go higher than the maximum achievable overlap? If so, is this an error, or have I misunderstood?”

This maximum overlap is a coarse theoretical estimate based on random patterns; thus, overlaps from our simulations can exceed it. We elaborate on this point in lines 178 and 604.

Minor point 6: “Top p. 6 – “because since” typo”

We thank the Reviewer for pointing this out.

Minor point 7: “p. 6 – “During retrieval, the network is asynchronously updated via Glauber dynamics (Amit et al., 1985). That is, at each simulation timestep, one neuron is randomly selected to be updated (Fig. 2C). If its total input from other neurons exceeds a threshold θ , then it is more likely to become active.” In this scheme, is there only further activation, or do active neurons also become randomly selected to possibly become inactivated (balancing the overall activation level)?”

Indeed, it is worth clarifying this point, and we now do so in line 150. Inactivation also occurs and it is more likely for subthreshold values of the total input.

Minor point 8: “

Bottom p. 6 to p. 7 – For myself as a reader, these paragraphs were incredibly helpful – I think the authors really hammer the main points home here well.”

We are pleased that our narrative was helpful to the Reviewer.

Minor point 9: “p. 10 – I very much appreciate this alternative prediction from the attention literature. It clarifies the need to resolve it via data, as they do below.”

We thank the Reviewer for this comment, and we now also place our predictions in the broader context of subtractive vs. divisive inhibition in response to Reviewer 1’s minor point 4 (lines 245 and 417).

Minor point 10: “Discussion: By my reading, Kowadlo et al. (2019, arXiv) & Antony et al. (2022, bioRxiv) present computational models that also argue for some generalization occurring in the PP (EC-CA3) pathway. The authors should consider how their model’s predictions support or differ from these papers.”

We appreciate the Reviewer’s help in pointing us to these very relevant papers. We discuss the relationship between their works and ours in lines 502 and 524. In brief, Antony et al. (2022) developed a hippocampal model along the lines of Schapiro et al. (2017) and Sučević & Schapiro (2022), which were discussed in our manuscript. Their model performs decontextualization, which is another form of generalization along with the concept learning that we study. Indeed, they demonstrate that lesioning the PP pathway decreases the network’s ability to associate cues with targets in the presence of additional contextual inputs that drift in time, but this lesion also decreases cue-target association with fixed contextual inputs. Thus, some ambiguity exists whether the PP pathway contributes specifically to generalization or aids associations in general. Kowadlo et al. (2019) developed a hippocampus model similar to ours in Fig. 2, but applied it to a machine learning task similar to ours in Fig. 8. With respect to both figures/sets of results, there are differences between our works. The Hopfield-like network for CA3 in their model only stores MF patterns. Moreover, they seek to improve multi-task learning by developing a new network architecture, whereas we seek to do so by developing a new loss function that can be applied to many different architectures.

Minor point 11: “(Addressing this point is optional.) My broadest question is how would these different theta phases playing different roles in both encoding and retrieval play out in the subjective encoding or retrieval of a memory? Do the authors believe a rodent or human can meaningfully alter (or pay attention to) the sparsity of their representations at the rate of a theta rhythm? If one were asked to recall the specific or conceptual associations to a given cue, would a control mechanism (say, from the prefrontal cortex) properly tune the theta rhythm in some way to accomplish this task? I think speculation along these lines would be helpful if the authors have insight, but if it is too speculative, it’s fine to omit.”

We thank the Reviewer for allowing us to entertain a more speculative and ambitious idea. We are intrigued by it without being aware of specific evidence in support or against it, and we mention it in line 483.

Minor point 12: “Methods: Line 527 – “expect” -> “except””

We thank the reviewer for pointing out this mistake.

Please do not hesitate to contact us if you have any further questions and/or comments.

Sincerely,

Louis Kang
Unit Leader, Neural Circuits and Computations Unit
RIKEN Center for Brain Science
louis.kang@riken.jp

Taro Toyozumi
Team Leader, Laboratory for Neural Computation and Adaptation
RIKEN Center for Brain Science
taro.toyoizumi@riken.jp

REVIEWER COMMENTS

Reviewer #1 (Remarks to the Author):

Kang and Toyozumi — NCOMMS-23-10468-T

Overview

I'd like to thank the authors for their efforts in responding to my previous comments. The revised manuscript is substantively improved, particularly with respect to clarity around the boundaries between the biological vs. AI-based aspects of the model designs and findings. Given its length and complexity, I still think this should ideally be two papers, but the improvements in the framing and narrative flow should allow this paper to stand on its own and potentially help bridge readers across computational neuroscience and AI research domains. The addition of Figure S3F supports the claim in the authors' response that the model parameters did not require fine-tuning. Moreover, various textual revisions and the addition of the Table 1 listing model parameters are very helpful for enhancing readability and comprehensibility of the models developed and investigated in the paper.

Major comment

Regarding the treatment of "grandmother cells" in response to "Minor point 2", I maintain that the authors' usage of the concept in the introductory text (lines 64–68 and the new Fig. 1C in the revised manuscript) is wrong, at odds with the field's understanding of the concept, and muddled with unrelated associations. The revised manuscript asserts that "remembering the details of an individual experience with your grandmother is a prototypical example of hippocampus-dependent episodic memory" (line 64; with "prototypical" added in revision). It is far from obvious why a "grandmother experience" is "prototypical" of episodic memory. The intent appears to be to set up the episodic→semantic transformation (i.e., "grandmother experience" → "grandmother (concept) cell") as the same kind of example→concept generalization process that is studied in the rest of the paper. However, the notion of "grandmother cell" as understood in the field is a *reductio ad absurdum* of the sparse coding hypothesis. The authors' response appears to assume that I was impugning the work of Quiroga et al. (2005, 2009) as denying distributed representations, which was not the case. Instead, my intention was to note that the authors were incorrectly associating Quiroga et al.'s work with the notion of "grandmother cell". Rodrigo Quiroga and colleagues even went so far as to publish a 2008 TICS opinion piece that explicitly separates their work from that notion, entitled "Sparse but not 'Grandmother-cell' coding in the medial temporal lobe" (<https://doi.org/10.1016/j.tics.2007.12.003>), in which they lay out the evidence-based

argument for the concern underlying my original comment: The grandmother cell "code" is maximally sparse and minimally (i.e., not) distributed. Thus, associating hippocampal episodic memory encoding or MTL semantic encoding with "grandmother cells" is equivalent to denying distributed coding, which the authors have claimed was not their intention. As I see it, lines 64–68 (starting with "For instance, ...") and Fig. 1C are confusing, misleading, and do not contribute to the scientific background or the findings of the paper, and thus should be replaced or removed.

Minor comments

- Line 106 (and elsewhere in the m.s.): instances of "postsynaptic density" should be rephrased, since that term already has a very specific biological meaning about synaptic structure.

- Line 128: the phrasing is difficult to parse and seems to imply direct cortical outputs from CA3, perhaps something like "...with the neocortical output pathway from CA3 to CA1 and deep layers of EC...".

- Line 130: Similar clarity issue here, e.g., suggest rephrasing "The true pathways involving CA1 are more complex and include..." as "The neuroanatomical connectivity of CA1 is more complex and includes..."

- Line 150: new text uses the word "inactivation", which already has a very specific biological meaning about synaptic ion channels; suggest using "silence" or similar word instead.

- I appreciate the new stacked bar charts in Fig. 5C, but it would be helpful if the left/right bars had x-axis labels, e.g., "example", "concept".

- Lines 249-250: new text "though subsequent implications...established" is not clear; what kind of implications are we talking about? The firing rate vs. theta phase relationship is well studied, so it help to understand specifically what remains to be established.

- Line 250: This newly added text is unclear: "Our model predicts that theta acts subtractively on neural activity in CA3". I'm confused by this sentence for several reasons: (1) In the model (e.g., as in line 417), theta just changes the threshold, so it's the threshold—not theta—that acts on neural activity; (2) Whether subtractive or division inhibition is implemented in a model is purely a factor of the model's design, so it's not clear how this could even constitute a prediction; (3) This is the final sentence of a paragraph with no clear sequelae; and (4) The prediction appears to obviate the earlier discussion

around the equivalent validity of divisive vs. subtractive effects of inhibition on place-field activity. Altogether, this sentence is perplexing and should be clarified, elaborated, or removed.

- Lines 468–486: This added discussion of STDP and plasticity rules is weird and doesn't clearly have a point. Temporal contiguity requirements are not about offsetting integration delays, they're about the complex sequential and correlational structures of neural inputs. There's a lot of language that anthropomorphizes neural circuits, like "perseverate" or "encourage". The argument treats "theta power" as both a dependent and independent variable, when methodologically it's a fairly abstract quantification of collective neural activity, so it's not clear if this discussion is presenting an argument for feedforward entrainment, lateral synchronization, collective emergence, or some other form of downward causation. Additionally, the last part (lines 483–486) about "overall inhibition level" (what does that mean?) and involuntary vs. voluntary control is confusing and out-of-place; episodic memory encoding is conventionally considered an automatic behavior.

Reviewer #2 (Remarks to the Author):

Thank you for addressing the reviewers' feedback and providing detailed responses to each point raised. I appreciate the effort made to address my concerns regarding the study. However, I still have some reservations regarding the first major point.

While I understand that defining the 'concept' qualitatively is challenging, I expected the authors to provide an explanation of how the mean image of each class can effectively represent the concept of a category. Although it is reasonable to utilize larger unsupervised deep neural network models like GAN or VAE to extract less variant features from a category in complex datasets such as Cifar10 or ImageNet, it is crucial to justify how these mean latent features truly represent the concept of a category. This justification is particularly important considering the argument made by the authors in the abstract: "...In this way, the model learns to transition between concept and example representations by controlling inhibitory tone..."

Additionally, as Reviewer 1 pointed out, there remains a gap between the neuroscience and AI/ML aspects in the paper. Instead of dividing this work into two separate papers, I would recommend that the authors focus on the first two parts of their research: the computational modeling and the hippocampal data analysis. They can then utilize AI/ML as an exemplar to demonstrate the model's extension.

Furthermore, I would like to suggest that the authors consider referencing the recent publication, "Assessments of dentate gyrus function: discoveries and debates" by Borzello et al. (2023), in order to

strengthen the validation of their ideas. This publication could provide additional support and context for their research.

Overall Recommendation Summation: Based on the authors' responses and the revisions made, I am inclined to recommend this revised paper for publication in Nature Communications. I believe the authors have adequately addressed the concerns raised during the review process. However, I would encourage the authors to consider the suggestions and points mentioned above to further enhance the clarity and coherence of the paper. I would also be happy to discuss any remaining questions or concerns with the reviewers and the editor, if necessary.

Reviewer #3 (Remarks to the Author):

The authors have done an excellent, thorough job with this revision. I believe this paper will spark great interest in both the neuroscience and machine learning fields. I have no more concerns. Bravo!

24 August 2023

Dear *Nature Communications* Editors and Reviewers:

Below we explain how our new manuscript has addressed all the Reviewers' new comments. Changes in the main text have been indicated in red, and all figure and line numberings invoked in this response refer to the revised version of the manuscript.

Reviewer 1

Major point 1: "Regarding the treatment of "grandmother cells" in response to "Minor point 2", I maintain that the authors' usage of the concept in the introductory text (lines 64–68 and the new Fig. 1C in the revised manuscript) is wrong, at odds with the field's understanding of the concept, and muddled with unrelated associations. The revised manuscript asserts that "remembering the details of an individual experience with your grandmother is a prototypical example of hippocampus-dependent episodic memory" (line 64; with "prototypical" added in revision). It is far from obvious why a "grandmother experience" is "prototypical" of episodic memory. The intent appears to be to set up the episodic-semantic transformation (i.e., "grandmother experience" -> "grandmother (concept) cell") as the same kind of example-concept generalization process that is studied in the rest of the paper. However, the notion of "grandmother cell" as understood in the field is a reductio ad absurdum of the sparse coding hypothesis. The authors' response appears to assume that I was impugning the work of Quiroga et al. (2005, 2009) as denying distributed representations, which was not the case. Instead, my intention was to note that the authors were incorrectly associating Quiroga et al.'s work with the notion of "grandmother cell". Rodrigo Quiroga and colleagues even went so far as to publish a 2008 TICS opinion piece that explicitly separates their work from that notion, entitled "Sparse but not 'Grandmother-cell' coding in the medial temporal lobe" (<https://doi.org/10.1016/j.tics.2007.12.003>), in which they lay out the evidence-based argument for the concern underlying my original comment: The grandmother cell "code" is maximally sparse and minimally (i.e., not) distributed. Thus, associating hippocampal episodic memory encoding or MTL semantic encoding with "grandmother cells" is equivalent to denying distributed coding, which the authors have claimed was not their intention. As I see it, lines 64–68 (starting with "For instance, ...") and Fig. 1C are confusing, misleading, and do not contribute to the scientific background or the findings of the paper, and thus should be replaced or removed."

We did not previously appreciate that the grandmother cell hypothesis implied response to invariant representations of only a single object or person and, thus, that the work of Quiroga and colleagues did not imply the grandmother cell hypothesis. We thank the Reviewer for alerting this to us, and we now cite explicit examples from the Scoville & Milner (1957) and the Quiroga et al. (2009) papers instead of invoking grandmother cells in lines 64–68. We have also replaced the grandmother character in Fig. 1C with abstract text that is more neutral. We have finally removed the term "prototypical."

Minor point 1: "Line 106 (and elsewhere in the m.s.): instances of "postsynaptic density" should be rephrased, since that term already has a very specific biological meaning about synaptic structure."

We have replaced this term with "postsynaptic pattern density."

Minor point 2: "Line 128: the phrasing is difficult to parse and seems to imply direct cortical outputs from CA3, perhaps something like "...with the neocortical output pathway from CA3 to CA1 and deep layers of EC..."."

We have made the suggested change.

Minor point 3: “Line 130: Similar clarity issue here, e.g., suggest rephrasing "The true pathways involving CA1 are more complex and include..." as "The neuroanatomical connectivity of CA1 is more complex and includes..."”

We have made the suggested change.

Minor point 4: “Line 150: new text uses the word "inactivation", which already has a very specific biological meaning about synaptic ion channels; suggest using "silence" or similar word instead.”

We have made the suggested change.

Minor point 5: “I appreciate the new stacked bar charts in Fig. 5C, but it would be helpful if the left/right bars had x-axis labels, e.g., "example", "concept".”

We have made the suggested change.

Minor point 6: “Lines 249-250: new text "though subsequent implications...established" is not clear; what kind of implications are we talking about? The firing rate vs. theta phase relationship is well studied, so it help to understand specifically what remains to be established.”

We meant to highlight that inhibition can cause either firing rates or membrane voltages to be subtractively or divisively modulated. As explained in Ferguson & Cardin, *Nat Rev Neurosci* (2020), under some assumptions, divisive modulation of membrane voltage can lead to subtractive modulation of firing rate. Thus, although membrane voltage was observed to be divisively modulated in hippocampus, the subsequent effect on firing rate cannot be concluded. We have decided to remove the references to membrane voltage modulation, because they do not directly pertain to the firing rate modulation that concerns Fig. 5E; thus, they may cause unnecessary confusion. We now explicitly state in line 262 that the subtractive and divisive types of inhibition reflected in Fig. 5E pertain to firing rates.

Minor point 7: “Line 250: This newly added text is unclear: "Our model predicts that theta acts subtractively on neural activity in CA3". I'm confused by this sentence for several reasons: (1) In the model (e.g., as in line 417), theta just changes the threshold, so it's the threshold—not theta—that acts on neural activity; (2) Whether subtractive or division inhibition is implemented in a model is purely a factor of the model's design, so it's not clear how this could even constitute a prediction; (3) This is the final sentence of a paragraph with no clear sequelae; and (4) The prediction appears to obviate the earlier discussion around the equivalent validity of divisive vs. subtractive effects of inhibition on place-field activity. Altogether, this sentence is perplexing and should be clarified, elaborated, or removed.”

We agree that this text is imprecise, and we have replaced it in line 264 with “We will now test whether experimental data reflect our model prediction of sharper place field tuning with higher spatial information during sparser theta phases, which would support a subtractive role of theta as an oscillating inhibitory threshold over a divisive one.” This addresses the four points of confusion: (1) We state that our prediction involves theta acting as an inhibitory threshold. (2) We agree that subtractive inhibition is a consequence of our model’s design, but the assertion that this model with this design applies to CA3 is a prediction that can be verified or refuted with experimental comparisons. In particular, we state the metric to be tested: the difference in spatial information between sparse and dense theta phases. (3) We now link this paragraph with the next one by explicitly stating that conducting these comparisons is our next task. (4) We meant to convey that subtractive and divisive inhibitory effects are equally valid as hypotheses, but the biological system could predominantly exhibit one or the other. We hope that our new passage, in addition to explicitly stating “hypothesis” in line 258, better reflects our intended meaning.

Minor point 8: “Lines 468–486: This added discussion of STDP and plasticity rules is weird and doesn't clearly have a point. Temporal contiguity requirements are not about offsetting integration delays, they're about the complex sequential and correlational structures of neural inputs. There's a lot of language that anthropomorphizes neural circuits, like "perseverate" or "encourage". The argument treats "theta power" as both a dependent and independent variable, when methodologically it's a fairly abstract quantification of collective neural activity, so it's not clear if this discussion is presenting an argument for feedforward entrainment, lateral synchronization, collective emergence, or some other form of downward causation. Additionally, the last part (lines 483–486) about "overall inhibition level" (what does that mean?) and involuntary vs. voluntary control is confusing and out-of-place; episodic memory encoding is conventionally considered an automatic behavior.”

Indeed, this section of the discussion goes beyond commonly agreed-upon knowledge and speculates upon potential ideas that may indeed be false. We added this section to address some questions by the Reviewers that have no definitive answer. Perhaps in our attempt to address these questions, we veered too far away from convention. In response, we choose to still address these topics, but we eliminate the most speculative components of our discussion. In doing so, we moved this trimmed paragraph to line 470, focused it around a central point, and rephrased the transition into the subsequent paragraph (line 480).

We can see how our discussion of STDP is speculative and out of place. We considered STDP in the context of Reviewer 2's suggestion to consider the consequences of a consistent temporal offset between MF and PP inputs. We have replaced this discussion of asymmetric STDP with a statement that temporal offsets may lead to nonlinear summation in the connectivity matrix in line 141.

We have removed the anthropomorphic language.

We intended to treat theta power as a downstream correlate of the inhibitory theta oscillation amplitude, which is possibly determined by the strength of medial septum inputs or the concentration of neuromodulators. We have clarified this direction of causation leading towards, not from, theta power. In other words, theta power is a dependent variable that has been observed to correlate with memory performance.

We have removed the discussion about involuntary vs. voluntary control, which Reviewer 3 also described as optional and speculative.

Reviewer 2

Major point 1: “While I understand that defining the 'concept' qualitatively is challenging, I expected the authors to provide an explanation of how the mean image of each class can effectively represent the concept of a category. Although it is reasonable to utilize larger unsupervised deep neural network models like GAN or VAE to extract less variant features from a category in complex datasets such as Cifar10 or ImageNet, it is crucial to justify how these mean latent features truly represent the concept of a category. This justification is particularly important considering the argument made by the authors in the abstract: "...In this way, the model learns to transition between concept and example representations by controlling inhibitory tone...”

We take this opportunity to elaborate upon our identification of mean images and target concept patterns with concept representations in line 190. We now reference ideas from both machine learning and cognitive science in which clustering is used as an algorithm to learn categories in an unsupervised fashion. Moreover, central features of these clusters can serve as prototypes for the categories. We then clarify that more sophisticated feature extraction networks, such as a VAE or a deep classifier, can be used to obtain these central features for more complicated datasets only if they can map the data into an encoding space in which they exhibit clusters between concepts. We hope this new text clarifies how more powerful feature extraction techniques can preserve our model's capabilities with more complicated datasets while acknowledging the limits of these techniques.

Minor point 1: “Additionally, as Reviewer 1 pointed out, there remains a gap between the neuroscience and AI/ML aspects in the paper. Instead of dividing this work into two separate papers, I would recommend that the authors focus on the first two parts of their research: the computational modeling and the hippocampal data analysis. They can then utilize AI/ML as an exemplar to demonstrate the model's extension.”

We appreciate the Reviewer’s suggestion to maintain the neuroscience and ML components of this work as one manuscript, which we have followed. We have added additional phrasing in lines 78, 363, and 516 to convey that the neuroscience component is a major contribution of our manuscript and that the ML component stems from it.

Minor point 2: “Furthermore, I would like to suggest that the authors consider referencing the recent publication, "Assessments of dentate gyrus function: discoveries and debates" by Borzello et al. (2023), in order to strengthen the validation of their ideas. This publication could provide additional support and context for their research.”

We thank the Reviewer for this reference, and we now include it in line 61.

Please do not hesitate to contact us if you have any further questions and/or comments.

Sincerely,

Louis Kang
Unit Leader, Neural Circuits and Computations Unit
RIKEN Center for Brain Science
louis.kang@riken.jp

Taro Toyozumi
Team Leader, Laboratory for Neural Computation and Adaptation
RIKEN Center for Brain Science
taro.toyoizumi@riken.jp

REVIEWERS' COMMENTS

Reviewer #1 (Remarks to the Author):

I thank the authors for the constructive consideration of my (and the other reviewer's) previous comments and for the significant efforts made to revise this manuscript. At this point, all of my concerns have been addressed and I believe that the result is an innovative, readable, and potentially impactful contribution to the emerging literature linking the neuroscience of learning and memory to artificial intelligence research. I support the publication of this article in its current form.

Reviewer #2 (Remarks to the Author):

The authors have answered my questions in the response letter and the revised paper. I appreciate the effort the authors made during the revision process. I have no more concerns!

22 November 2023

Dear *Nature Communications* Editors:

We are pleased that all the reviewers have endorsed publication of the current version of the manuscript. Please do not hesitate to contact us if you have any further questions and/or comments.

Sincerely,

Louis Kang
Unit Leader, Neural Circuits and Computations Unit
RIKEN Center for Brain Science
louis.kang@riken.jp

Taro Toyozumi
Team Leader, Laboratory for Neural Computation and Adaptation
RIKEN Center for Brain Science
taro.toyoizumi@riken.jp